# 🏆 COPA: Comparing the incomparable
# in multi-objective model evaluation

## Abstract

In machine learning (ML), we often need to choose one among hundreds of trained ML models at hand, based on various objectives such as accuracy, robustness, fairness or scalability. However, it is often unclear how to *compare*, *aggregate* and, ultimately, *trade-off* these objectives, making it a time-consuming task that requires expert knowledge, as objectives may be measured in different units and scales. In this work, we investigate *how* objectives can be automatically normalized and aggregated to systematically help the user navigate their Pareto front. To this end, we make incomparable objectives comparable using their cumulative functions, approximated by their relative rankings. As a result, our proposed approach, COPA, can aggregate them while matching user-specific preferences, allowing practitioners to meaningfully navigate and search for models in the Pareto front. We demonstrate the potential impact of COPA in both model selection and benchmarking tasks across diverse ML areas such as fair ML, domain generalization, AutoML and foundation models, where classical ways to normalize and aggregate objectives fall short.

## 1 Introduction

In all steps of a ML pipeline, from model development to deployment, we often need to *compare and select one trained model among a population according to different objectives.* Even for a simple classification task, model selection often involves comparing trained classifiers that trade-off objectives such as accuracy, sensitivity or specificity (Japkowicz & Shah, 2011), and realistic settings often require benchmarking lots of models in terms of diverse objectives such as robustness (Yuan et al., 2023), fairness (Huang et al., 2023), and $CO_2$ footprint (Coignion et al., 2024; Luccioni et al., 2023). Unfortunately, it is unclear *how to systematically compare and select a model in terms of multiple objectives among a given population.*

Furthermore, *different users have different preferences.* For example, imagine a user who wants to use a trained large language model (LLM) from the *Open LLM Leaderboard* (Fourrier et al., 2024) to solve a mildly challenging task. To this end, the user wants an LLM that performs relatively well without unnecessarily large $CO_2$ footprints. In total, there are 2148 available LLMs and 7 objectives, 6 performance benchmarks and inference cost. Among these, 487 present non-trivial trade-offs, i.e., for every pair, one is better in an objective but worse in another. How should they compare these models to make a decision? Should they manually inspect all 487? And if later they require a more robust model: Should they start from scratch?

Most remarkably, this ambiguity is unavoidable and exists as soon as we have several objectives. In other words, when we make a decision, *we always choose a way of comparing and selecting the model best aligned with our preferences*, independently of whether we make this choice explicit or not. Moreover, the example above highlights two main obstacles in this decision process. Namely:

**L1.** Objectives with different semantics and domains are not directly *comparable*, e.g., average score and $CO_2$ cost in Fig. 1, and thus cannot be properly aggregated nor traded-off. In physics, this would be akin to comparing values measured in different units, e.g., meters and grams.

**L2.** When the number of objectives grow, humans have a difficult time translating their preferences into a concrete decision, as the number of choices quickly becomes overwhelming (487 in our example).

While **L2** motivates the need of tools to help the user navigate the Pareto front (i.e., the optimal trade-offs), **L1** hinders these tools to reliably reflect the user preferences. This is illustrated in Fig. 1 which, coming back

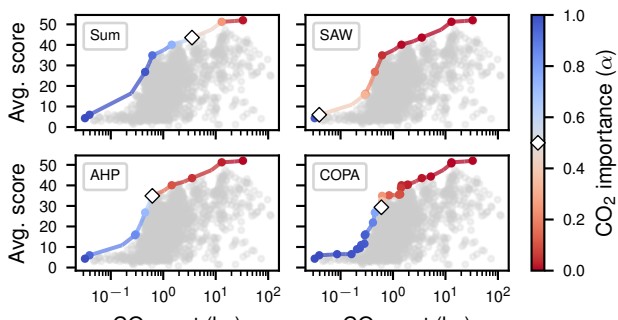

Figure 1: **COPA meaningfully navigates the performance-cost trade-offs in the Open LLM Leaderboard**, sensibly mapping the importance of $CO_2$ cost to the Pareto front (left). In contrast, existing approaches such as AHP and SAW (see §A) are either biased toward one of the objectives or find few solutions (colored dots). This is reflected in the retrieved LLMs (right) where COPA maps $\alpha = 1/2$ to a top-18 % model for both objectives.

to the previous example, takes 4 different methods and colors the Pareto front according to the importance given to $CO_2$ cost (left), retrieving the LLM corresponding to no preference (right). Throughout this work, we consider a good mapping one which *uniformly* maps our objective preferences (importance values going from zero to one) to the Pareto front, thus easing its navigation through our preferences. As such, simple weighted sums, Sum and SAW (MacCrimmon, 1968), lead to biased solutions either toward the score or $CO_2$ cost. In contrast, more involved solutions from the decision-making literature like AHP (Saaty, 1990; 1977) provide a well-balanced LLM for $\alpha = 1/2$ (i.e., for equal preference between average score and $CO_2$ cost), but only *explore* a small subset of models in the Pareto Front as we change $\alpha$ (colored dots). In practice, this issue is overcome by using heuristics to normalize all objectives (Nazabal et al., 2020; Caruana & Niculescu-Mizil, 2004), e.g., the DecodingTrust benchmark (Wang et al., 2023), introduces 8 rules, one per objective, to normalize the different objectives. (see §B.6). In view of the lack of general tools to systematically *compare, aggregate and, ultimately, trade-off objectives*, we propose **COPA** 🏆 (*cumulative-based optimization of the Pareto front*, introduced in §3), a simple approach that better maps the user preferences into the Pareto front.

We find these challenges ubiquitous in model evaluation and selection tasks, as *they appear each time we attempt to compare two models over multiple objectives*, possibly biasing any conclusion draw from said comparisons. This is the case, for example, when comparing the performance of different models in multitask learning (MTL) or domain generalization research (Navon et al., 2022; Ramé et al., 2022), where the model is *implicitly* expected to work 'well' on all tasks or domains; in fair classification (Zafar et al., 2017), where it is often unclear what is an acceptable fairness-accuracy trade-off for deployment; or in AutoML (Gijsbers et al., 2024), where dozens of frameworks are compared on hundreds of objectives.

**Our contributions are as follows:** First, we motivate and *discuss* the problem of incomparability in multi-objective ML evaluation, shedding light on why previous approaches fail and how it affects us (§2). Next, we introduce **COPA** 🏆, a simple tool to *help practitioners meaningfully navigate the Pareto front*, and thus compare and select models that reflect their preferences (§3). To this end, COPA uses **i)** a normalization function that *universally* makes all objectives comparable via their cumulative distribution functions; and **ii)** a simple criterion function with two interpretable parameters controlling the aggregation and importance of each objective. Then, after placing COPA in the context of related work (§4), we demonstrate its potential impact in diverse and timely areas such as MTL, domain generalization, fair ML, AutoML benchmarking and LLM selection (§5). As Fig. 1 exemplifies, COPA enables the thorough exploration of the Pareto front as a function of the user preferences, here controlled by $\alpha$, where, e.g., a deployer equally interested in the performance and $CO_2$ footprint of the LLM could use COPA with $\alpha = 1/2$ to pick a model in the middle of the Pareto front, ranked top-18 % for both objectives (last row in Fig. 1, right).

## 2 Problem statement

We are given a population of trained models $\mathcal{H}$, where each model $h \in \mathcal{H}$ is associated to a vector of $K$ metrics assessing its performance with respect to different evaluation objectives. In addition, we assume each objective to be a continuous random variable for which we have sampled observations in $\mathcal{H}$.

Without loss of generality, we assume that each individual objective has to be *minimized*, and we can thus frame the problem as a multi-objective optimization (MOO) problem of the following form:

$$\min_{h \in \mathcal{H}} \mathbf{y}(h) := [y_1(h), y_2(h), \ldots, y_K(h)], \tag{1}$$

where $\mathbf{y}(h)$ is the objective vector of model $h$, and $y_k(h)$ its performance on the $k$-th objective. When it is clear from the context, we will omit the argument $h$ and write $\mathbf{y}$ and $y_k$ instead.

**When do we minimize a vector?** In this work, we adopt a loose sense of the min operator in Eq. 1. Besides those use cases where the minimization problem is explicitly solved as in, e.g., model selection, we also consider settings where all models are given and it is the **decision maker** (DM) the one comparing these models in order to look for a best (i.e., minimal) model. This interpretation allows us to show in §5 diverse use cases which we can interpret as in Eq. 1, e.g., comparative model analysis (§5.3) where, given a table of models and their objectives, one draws conclusions on which models perform better than others.

**How can we minimize a vector?** A fundamental problem of Eq. 1 is that *minimizing the vector* $\mathbf{y}$ *is not well-defined*, as there is no canonical total order in high dimensions. Hence, two models could yield objective vectors where one is not always better than the other for all objectives. In the MOO literature, the set of optimal trade-off solutions is known as the *Pareto front* and, more formally, an objective vector $\mathbf{y}^*$ is in the Pareto front (and called *Pareto-optimal*) if there exists no other feasible vector $\mathbf{y}$ such that $y_k \leq y_k^*$ for all $k \in \{1, 2, \ldots, K\}$, and $y_k < y_k^*$ for at least one of the objectives.

Eventually, the DM needs to navigate the Pareto front and select one given model.[1] That is, the DM needs to specify a total order in Eq. 1. This is *unavoidable and intrinsic to the problem nature*. There are two options: **i)** take a total order directly in $\mathbb{R}^K$, e.g., the lexicographic order where $\mathbf{y} < \mathbf{y}^*$ iff $y_k < y_k^*$ and $y_i = y_i^* \ \forall i < k$; or **ii)** define a **criterion function** $C \colon \mathbb{R}^K \to \mathbb{R}$ to rewrite Eq. 1 as a scalar-valued problem:

$$\min_{h \in \mathcal{H}} \quad C(\mathbf{y}(h)). \tag{2}$$

One remarkable example of the latter is the *global-criterion method* (Zeleny, 1973) which maps DM preferences to the problem geometry by interpreting Eq. 2 as selecting the model closest to the *ideal* one, i.e.,

$$\min_{h \in \mathcal{H}} \quad \|\mathbf{y}(h) - \mathbf{y}^{\text{ideal}}\|_*, \tag{3}$$

where $\mathbf{y}^{\text{ideal}}$ is the ideal solution, $\mathbf{y}^{\text{ideal}} := [\min_h y_1, \min_h y_2, \ldots, \min_h y_K]$, and $\|\cdot\|_*$ is typically a $p$-norm. However, naively solving Eq. 3 (and, more generally, Eq. 2) is well-known in the MOO literature to be sensitive to the scaling of the objectives (Branke et al., 2008) (recall **L1** in §1), and thus hinder us from properly accounting for the DM preferences (**L2**) by picking the right norm. In this work, we argue that the criterion function $C$ should fulfill the following desiderata:

**D1.** Reflect the DM preferences, translating their model expectations into an optimization problem.

**D2.** Provide a simple way to tune its parameters to meaningfully explore the Pareto front.

**When are objectives incomparable?** Similar to dimensional analysis in physics (see e.g., Barenblatt (1987)), which argues that we cannot combine incommensurable quantities (e.g., kilograms and meters), we argue that a second fundamental issue that we face in Eq. 2 is what we call **semantic incomparability**, i.e., whether it is sensible to compare (and thus aggregate) the values of two different objectives.

---

[1]Note that, when we plot the Pareto front as in Fig. 1, the linear interpolation between models (colored dots) only serves visualization purposes, i.e., we do not interpolate between models.

In general, if objectives differ in their semantics they are hardly comparable, e.g.: despite both accuracy and ROC AUC lying in the unit interval, it does not make immediate sense to compare their values. There are, however, more nuanced aspects to it. To illustrate these, Fig. 2 presents a synthetic Pareto front from §5.1 where both objectives quantify regression errors in significantly different domains, namely, within the intervals $[0, 0.2]$ and $[0.5, 3.0]$. To navigate the Pareto front, we look at the MOO literature and formulate Eq. 3 as a weighted Tchebycheff problem (Bowman, 1976) of the form

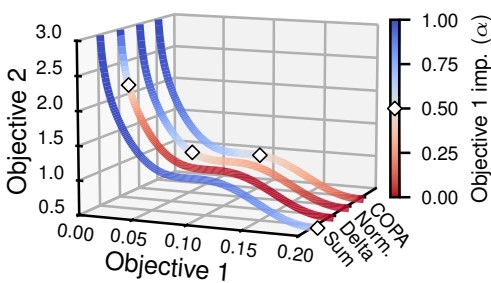

Figure 2: As we apply different normalization functions to the synthetic Pareto front from §5.1 to solve Eq. 7, only COPA meaningfully navigates it as we change $\alpha$.

$$\min_{h \in \mathcal{H}} \ \max \left\{ \alpha |\mathrm{y}_1|, \ (1 - \alpha)|\mathrm{y}_2| \right\} , \tag{4}$$

which solves Eq. 3 with $C$ as the $\infty$-norm, weighted by $\alpha \in [0, 1]$. Intuitively, Eq. 4 looks for robust solutions that account for the importance of solving one objective over the other, seemingly satisfying our desiderata, **D1-2**. However, its naive application over the original objectives clearly shows how we can bias the problem in favor of Objective 2, as it can be seen in Fig. 2: for any given preference $\alpha$ smaller than 0.75, Eq. 4 yields a solution which *completely ignores Objective 1 performance.*

***How* can we make objectives comparable?** As just argued, semantic incomparability can hamper *a well-designed criterion function* from meaningfully exploring the Pareto front. In the MOO literature, this has been typically addressed by applying *component-wise transformations* to the objectives to normalize them (Miettinen, 1999), turning Eq. 2 into

$$\min_{h \in \mathcal{H}} \ C(\boldsymbol{\phi}(\mathbf{y})) \coloneqq C \left( [\phi_1(\mathrm{y}_1), \ldots, \phi_K(\mathrm{y}_K)] \right) . \tag{5}$$

Two classic examples of these transformations are

$$\Delta_k(\mathrm{y}_k) \coloneqq \frac{\mathrm{y}_k - \mathrm{y}_k^{\mathrm{ideal}}}{\mathrm{y}_k^{\mathrm{ideal}}} , \quad \mathrm{norm}_k(\mathrm{y}_k) \coloneqq \frac{\mathrm{y}_k - \mathrm{y}_k^{\mathrm{ideal}}}{\mathrm{y}_k^{\mathrm{nadir}} - \mathrm{y}_k^{\mathrm{ideal}}} , \tag{6}$$

where $\mathrm{y}_k^{\mathrm{nadir}} \coloneqq [\max_h \mathrm{y}_1, \max_h \mathrm{y}_2, \ldots, \max_h \mathrm{y}_K]$ is the worst plausible solution. Other approaches, such as SAW (MacCrimmon, 1968) and AHP (Saaty, 1990; 1977), use max-normalization and spectral decomposition instead, see §A. Intuitively, $\Delta_k$ represents the relative difference to the ideal, while $\mathrm{norm}_k$ rescales the objective to lie in the unit interval. It is worth noting that prior works extensively used $\Delta_k$, often replacing $\mathrm{y}_k^{\mathrm{ideal}}$ with a reference vector, as computing it can be challenging (Miettinen, 1999; Maninis et al., 2019; Liu et al., 2023). Back to our example, we now want to solve

$$\min_{h \in \mathcal{H}} \ \max \left\{ \alpha |\phi_1(\mathrm{y}_1)|, \ (1 - \alpha)|\phi_2(\mathrm{y}_2)| \right\} . \tag{7}$$

By testing the $\phi_k$ defined in Eq. 6, we can understand why they fail to make objectives comparable. We find in Fig. 2 that: **i)** $\Delta_k$ biases the problem toward the first objective, since $\min_h \mathrm{y}_1 \approx 0$; and **ii)** $\mathrm{norm}_k$ alleviates the issue, as the denominator is now bigger than the numerator, yet distribution differences (that of $\mathrm{y}_2$ being more heavy-tailed) still bias the optimization toward the first objective. Instead, we seek to explore the Pareto front making a more meaningful use of $\alpha$, spreading it uniformly along the curve.

The *main goal* of $\phi_k \colon \mathbb{R} \to \mathbb{R}$ is thus to make the objectives semantically comparable, so that we can seamlessly aggregate them with the criterion function C. To this end, we argue that the functions $\phi_k$ should be:

**D3.** Objective-agnostic, so that we can normalize any objective irrespectively of its specific nature.

**D4.** Order-preserving (i.e., strictly increasing), so that it preserves Pareto-optimality.

In summary, to meaningfully explore the Pareto front, it is important to design a criterion function $C$ that translates well DM preferences into an optimization problem (**D1-2**), and a normalization function $\phi$ that makes objectives semantically comparable (**D3-4**) to not jeopardize the former. These desiderata will blend in COPA, as we discuss in the next section. In the synthetic experiment above, COPA maps the value $\alpha = 1/2$,

which turns Eq. 7 into a robust min-max problem (Verdu & Poor, 1984), to the flat region of the curve in Fig. 2, matching the intuition of what a robust solution should represent.

## 3 Methodology

Next, we introduce the proposed normalization and criterion functions fulfilling the desiderata **D1-4** described in §2. We refer to the problem resulting of solving Eq. 5 with the proposed functions as *cumulative-based optimization of the Pareto front* or, in short, **COPA ♔**.

### 3.1 Designing a universal normalization function

We argued in §2 that the normalization function $\phi$ should fulfill desiderata **D3-4**, i.e., it should make any objectives semantically comparable while preserving their Pareto-optimality. Taking a probabilistic perspective (recall that $y_k$ is a continuous random variable by assumption, §2), we propose to design $\phi$ such that the resulting variables are all equally distributed and, w.l.o.g., uniformly distributed in the unit interval. That is, we propose to use $\mathbf{u} \coloneqq [u_1, u_2, \dots, u_K]$ instead of $\mathbf{y}$, where

$$u_k \coloneqq F_k(y_k) \sim \mathcal{U}(0, 1) \quad \forall k \in \{1, 2, \dots, K\}, \tag{8}$$

and $\phi_k = F_k$ is the marginal cumulative distribution function (CDF) of the $k$-th objective. This transformation is indeed known in statistics as the probability integral transform (Casella & Berger, 2021, Thm. 2.1.10), and $u_k$ is guaranteed to follow a standard uniform distribution if $y_k$ is continuous.

More remarkably, Eq. 8 makes all criterion functions *marginal-distribution-free* in the sense of Kendall & Sundrum (1953), i.e., it strips away all individual properties of the marginal distribution (e.g., their domain) of any given objective (**D3**). We also note that normalizing random variables this way is one of the fundamental building stones of copulae in statistics (Sklar, 1959; Geenens, 2024), which ensures that copula functions exclusively learn the relationship across the random variables they model.

**How can we interpret the values of u?** One important advantage of using $\mathbf{u}$ in place of $\mathbf{y}$ in Eq. 5 is that it provides a *common framework* to think about all objectives, since all their values all are now framed as *elements within a population*. In practice, this means that the DM has a common language to express the expectations place on the model, e.g., for all objectives $u = 1/2$ corresponds to the *the median value* (i.e., top-50 %), which divides $\mathcal{H}$ into two *halves* comprising the best and worst performing models.

There still exists, however, one caveat we need to address: We have no access to the marginal CDF of each objective, but only to samples of the joint distribution provided in $\mathcal{H}$.

### 3.2 Rankings as finite-sample approximations

While we have no access to the CDFs, we have samples from the joint distribution over the objectives, i.e., over, $p([y_1, y_2, \dots, y_K])$. Hence, we can consider each model $h \in \mathcal{H}$ as a sample from the joint distribution and, by looking at each objective individually, as a sample from the marginal distributions.

Let us now focus on one objective, and drop their subindex in the following to ease notation. Say that we have $|\mathcal{H}| = N$ i.i.d. realizations of the objective, i.e., $\{y_1, y_2, \dots, y_N\} \overset{\text{i.i.d.}}{\sim} P$. Then, we can approximate Eq. 8 for the $i$-th sample, $u_i = F(y_i)$, by computing its order statistic, i.e., the random variable representing its relative ranking within the population, $R(i) \coloneqq \sum_{j=1}^{N} [y_j < y_i]$, where Iverson brackets denote the indicator function. Now, since the *empirical CDF* is defined as the fraction of samples smaller than its input, it is direct to show that

$$\hat{u}_i = \hat{F}(i) \coloneqq \frac{1}{N} \sum_{j=1}^{N} [y_j < y_i] = \frac{1}{N} R(i), \tag{9}$$

which is known as the *rank transform* (Conover, 2012) and holds the following (Vaart, 1998, Chapter 19):

**Proposition 3.1.** $\hat{u}_i$ *is an unbiased estimator of the CDF at* $y_i$, $u_i = F(y_i)$, *with variance* $u_i(1 - u_i)/N$. *Therefore, the variance of* $\hat{u}_i$ *decreases linearly with* $N$, *and has a maximum value of* $0.25/N$ *at the median.*

*Proof.* First, note that $[\,y_j < y_i\,] \sim \mathrm{Bern}(u_i)$. Then, we have $R(i) \sim \mathrm{Bin}(N, u_i)$ with mean $N u_i$ and variance $N u_i (1 - u_i)$. Hence, $\hat{u}_i$ has mean $\frac{1}{N} \mathbb{E}[R(i)] = u_i$, and variance $\frac{1}{N^2} \mathbb{V}[R(i)] = u_i(1 - u_i)/N$ which, by taking derivatives w.r.t. $u_i$, $\partial_{u_i} \mathbb{V}[\hat{u}_i] = 1 - 2u_i = 0 \Rightarrow u_i = 1/2$, which is a maximum since $\partial^2_{u_i} \mathbb{V}[1/2] < 0$ . $\qquad\square$

In other words, we can use the relative rankings of each objective to build an unbiased estimator of the CDF, $\hat{u}_i$, whose variance rapidly decreases as we increase the size of $\mathcal{H}$, i.e., $\mathbb{V}[\hat{u}_i] \to 0$ as $N \to \infty$, or rank extreme values, e.g., top models. This can be observed in the inset figure. In fact, it is known to be a consistent estimator (Tucker, 1959) with uniform convergence (Dvoretzky et al., 1956). Note also that relative rankings are strictly increasing: If $y_i < y_j$, then $\hat{F}(y_i) < \hat{F}(y_j)$ for any $\mathcal{H}$ containing both samples (**D4**). While this is an approximation of the true CDF, it works egregiously well in our experiments (§5). Finally, note that this transformation is meant to ease inter-objective computations in Eq. 5 by normalizing all objectives, and thus we can (and *should*) use the original objective values of $y_k$ to perform intra-objective comparisons and decisions, e.g., looking at the marginals to find phase transitions.

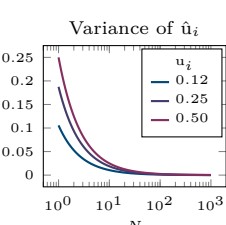

### 3.3 Incorporating preferences into the optimization

Assume we can effectively make objectives comparable, we can now focus on *simple* criterion functions that translate DM preferences into an optimization problem (**D1**). To do so, we start by looking back at global criterion methods, since plugging in our transformation $\mathbf{u} = F(\mathbf{y})$ simplifies the problem in Eq. 3 to $\min_h \|\mathbf{u}\|_*$ as the ideal point becomes the origin, i.e., $\mathbf{u}^{\mathrm{ideal}} = \mathbf{0}$. Then, by using the approximation described in §3.2, the problem becomes a simple finite search of the form

$$\min_{i \in \{1,2,\ldots,N\}} \ \|\hat{\mathbf{u}}_i\|_* \,. \tag{10}$$

That is, we have reduced our problem to finding the model whose ranking vector has the smallest norm. Using this *marginal-free global-criterion method*, mapping the DM preferences now boils down to selecting an appropriate norm for the problem in Eq. 10. To this end, we propose to use as criterion function $C$ a norm with parameters $p \geq 1$ and $\boldsymbol{\omega} \in \mathbb{R}^K_+$, where $\sum_k \omega_k = 1$, defined as

$$\|\mathbf{u}\|_{p,\boldsymbol{\omega}} := \left( \sum_{k=1}^{K} |\omega_k u_k|^p \right)^{1/p} , \tag{11}$$

This norm can be interpreted as a regular $p$-norm on a space with coordinates scaled by $\boldsymbol{\omega}$. More remarkably, note that Eq. 11 *is different* from the usual weighted $p$-norm, as the weights are *inside* the absolute value. With this parametrization, we can go from a weighted sum ($p = 1$) to a weighted Tchebycheff problem ($p = \infty$) (Bowman, 1976). Numerically, since the values of $u_k$ lie in the unit interval, a regular weighted p-norm would often make them vanish too quickly, as we empirically demonstrate in Fig. B.3.

**How can we interpret the parameters?** Fortunately, the parameters of Eq. 11, $p$ and $\boldsymbol{\omega}$, provide a simple and interpretable way for the DM to navigate the Pareto front (**D2**). Regarding $\boldsymbol{\omega}$, as we apply them in Eq. 11 *before* taking the power, we can interpret $\boldsymbol{\omega}$ in terms of ratio trade-offs. E.g., if we had two objectives and $\boldsymbol{\omega} = [0.75, 0.25]$, then equating the weighted objectives we see that minimizing the first objective to a value of $u_1$ is worth the same as minimizing the second objective to a value of $u_2 = \omega_1/\omega_2 u_1 = 3u_1$, i.e., $u_1$ is three times more important than $u_2$. Combining this interpretation with that of $\mathbf{u}$ in §3.1, we could say, e.g., that we value being in the top-25 % for the first objective the same as being in the top-75 % for the second one.

Geometrically, we can interpret $p$ using the same intuition as in ML regularization (Goodfellow et al., 2016): The models selected in Eq. 10 are those first intersecting an ever-expanding $p$-ball centered at the origin, whose shape depends on $p$. Therefore, higher values of $p$ lead to more uniform objective vectors (since individually bad results are penalized more heavily), while smaller values are less sensitive to changes in $\boldsymbol{\omega}$ but lead instead to more unbalanced solutions. This is concordance with the specific interpretations for values of $p$, e.g.: $p = 1$ is the average rank; $p = 2$ is the Euclidean distance; and $p = \infty$ turns Eq. 10 into a min-max problem, typically used to formulate robust optimization problems (Verdu & Poor, 1984).

**Does Eq. 11 enjoy theoretical guarantees?** Given the similarity with weighted p-norms, we can leverage existing results from the MOO literature and adapt them to Eq. 11. As a result, e.g., we know that the solutions found using Eq. 11 with $1 \leq p < \infty$ are always Pareto-optimal (Miettinen, 1999, Thm. 3.4.1), yet it might not reach all optima (indeed, for $p = 1$ it only reaches those in the convex hull). Similarly, since $p = \infty$ reduces Eq. 10 to a weighted Tchebycheff problem, we know that it reaches any Pareto-optimal solution (Miettinen, 1999, Thm. 3.4.5), but can also find weakly optimal ones.

In practice, we find that using a weighted Tchebycheff problem ($p = \infty$) is a good option when we have a large sample budget for the weights $\boldsymbol{\omega}$, as it can reach any Pareto-optimal point at the expense of being sensitive to changes in $\boldsymbol{\omega}$. Instead, if the DM is more relaxed about the model found or the exploration budget is limited, we suggest setting $p$ based on the level of robustness desired (smaller $p$ leading to higher tolerance to individual bad performance), and $\boldsymbol{\omega}$ based on the importance of solving each objective.

**Time complexity.** To utilize COPA, we need to normalize each objective using the finite-sample approximation in Eq. 9 and then compute the criterion function in Eq. 11 with the given user preferences. As a result, the time complexity of COPA amounts to sorting $K$ objectives and then aggregating their values, i.e., $\mathcal{O}(KN \log N)$, making COPA extremely lightweight to compute in practice.

## 4 Related work

In this section, we explore the relation of COPA with prior works in ML and other research fields.

**Relation with other sciences.** As mentioned in §1, our notion of semantically incomparability is similar in spirit to that of incommensurability in dimensional analysis in physics (Barenblatt, 1987). Similarly, relative rankings have been previously explored to make better comparisons in microeconomics (Piggins, 2019), MOO (Kukkonen & Lampinen, 2007), and statistics, designing methods that avoid the normality assumption, e.g., the Friedman test (Friedman, 1937), Wilcoxon signed-rank test (Wilcoxon, 1945), or Kendall's $\tau$ coefficient (Kendall, 1938). Finally, as mentioned in §2, copulae are notoriously known for exploiting the probability integral transform to become marginal-distribution-free (Geenens, 2024), and the proposed criterion functions share similarities with weighted $L_p$-problems in MOO (Miettinen, 1999).

**Related AI works.** Multiple works have been proposed that relate or are subsumed by COPA. For example, in multi-criteria decision making Fernando et al. (2011) proposed to use an average of rankings to aggregate objectives (COPA with $p = 1$). Similarly, Yamada et al. (2024) proposed a Tchebycheff formulation over rankings (COPA with $p = \infty$) to select a subset of the population in evolutionary algorithms, yet they use only Pareto-optimal points, disallowing some uses cases later explored in §5. In Bayesian optimization, Park et al. (2024) learns to approximate the joint CDF with a copula to recover a partial order in MOO problems. Another line of related works are those that attempt to learn the Pareto front either for model merging (Li et al., 2024; Chen & Kwok, 2024) or a posteriori MOO methods (Zhong et al., 2024). However, these methods ignore semantic incomparability as they use the original objectives. Lastly, ROC curves (Flach, 2010) provide an interesting connection, as their axes can be understood as the CDFs of the target classes (Hand, 2009).

**Potential impact.** Our work highlights a number of use cases (see §§2 and 5) for which semantic incomparability is underexplored, and often ignored, offering a simple and general solution to overcome it. As a result, COPA can benefit many works and applications in ML, e.g., all prior works proposing ad hoc ways to normalize and aggregate objectives (Nazabal et al., 2020; Wang et al., 2023; Shamsian et al., 2025). Furthermore, COPA offers a principled approach to evaluate ML models and subsumes initial approaches proposed in areas such as MTL (Navon et al., 2022; Liu et al., 2023) or domain generalization (Ramé et al., 2022), where rank averages are used to aggregate objectives. However, it is still common in these fields to use the average of $\Delta_k$-normalized objectives (see Eq. 6) for evaluation (Liu et al., 2021; Wang et al., 2025; Ban et al., 2025). More generally, COPA can benefit any field in which there are multiple objectives with which to compare different models, e.g., in fair ML (Martínez et al., 2020), federated learning (Kairouz et al., 2021), probabilistic ML (Javaloy et al., 2022) or multimodal learning (Baltrušaitis et al., 2018).

## 5 COPA in action

In this section, we motivate the use of COPA by showing a range of practical scenarios which would benefit from its adoption. We present additional details and results for all experiments in §B.

### 5.1 Synthetic evaluation

First, we consider a synthetic front given by $y_2 = 0.25\cos(39y_1^{0.85}) - \log(y_1) - 0.46$ and $y_1 \sim \mathcal{U}(0.02, 0.2)$. We obtain as a result a non-convex Pareto front with a flat area around $y_1 = 0.1$, and two objectives with significantly different distributions.

**Does $p$ match our intuitions?** We corroborate the insights from §3.3 by showing in Fig. 3 the distribution of solutions found taking different values of $p$ and $\alpha$. First, note that $u_1 \approx 1 - u_2$ since the front is *strictly* increasing except in $[0.083, 0.091]$. As a result, we find that $p = 1$ almost exclusively finds solutions on the extrema, i.e., it is insensitive to $\boldsymbol{\omega} = [\alpha, 1 - \alpha]$. When we increase $p$, the distribution of solutions better spreads along the front and, as the $p$-balls become more squared, COPA becomes more sensitive to $\boldsymbol{\omega}$

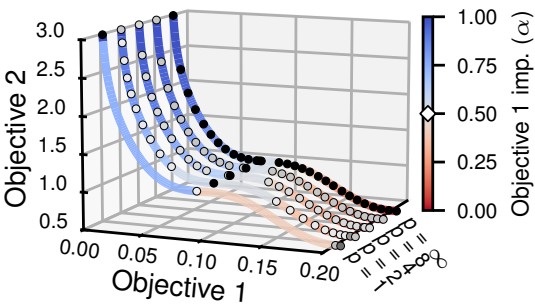

Figure 3: Distribution of solutions (circles) found for different values of $p$ as we sweep over values of $\alpha$. The darkness of the circles represents the number of times they were selected by changing $\alpha$.

and we thus have finer control on the solution found by tuning $\alpha$. It is important to stress, however, that the finer control of $p = \infty$ comes at a cost: if $K$ is large, properly tuning $\boldsymbol{\omega}$ could prove challenging.

### 5.2 Case 1: Model selection

Next, we explore how the weighted norm proposed in §3.3, Eq. 11, can helps us explore the Pareto front more meaningfully, i.e., by sensibly mapping DM preferences to the optimization problem in Eq. 5.

**1. The performance-emissions trade-off.** Despite LLMs recently showing outstanding performance (Naveed et al., 2023), their $CO_2$ footprint is a concern that needs to be taken into account (Coignion et al., 2024). Next, we show how practitioners can leverage COPA to better navigate this crucial trade-off.

We gather the results of 2148 LLMs from the Open LLM Leaderboard (Fourrier et al., 2024) and take as objectives their inference $CO_2$ cost and performance on 6 datasets: IFEval (Zhou et al., 2023), BBH (Suzgun et al., 2023), MATH (Hendrycks et al., 2021), GPQA (Rein et al., 2023), MuSR (Sprague et al., 2024), and MMLU-Pro (Wang et al., 2024). Then, we use COPA with $p = \infty$ to select an LLM, changing the importance given to their $CO_2$ footprint, $\alpha$, and setting $\boldsymbol{\omega} := [\alpha, (1-\alpha)/6, \ldots, (1-\alpha)/6]$.

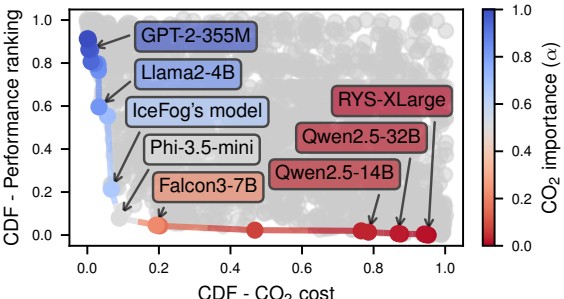

Figure 4: **COPA helps us meaningfully explore the Pareto front of the Open LLM Leaderboard** (Fourrier et al., 2024). We use $p = \infty$, 7 objectives, and highlight some selected models as we change the value of $\alpha$.

Complementing the discussion in §1, Fig. 4 highlights some LLMs by COPA, where we plot the CDFs of the $CO_2$ cost and the $\infty$-norm of all other objectives. Quantitative results can be found in §B.2.1. We observe that COPA helps us meaningfully explore the Pareto front, with the values of $\alpha$ being uniformly spread-out along the front. Moreover, not only can we sensibly explore the LLM space, but COPA enables interpreting these models in terms of the original objectives *and* the population they live in. For example, we can say that GPT-2 is Pareto-optimal as it consumes the least, but it only achieves a 6% average performance score, or that Phi-3.5-mini is a top-10% model in both aspects, consuming 0.53 kg of $CO_2$ vs. the 13 kg consumed by the best-performing model.

**2. The fairness-accuracy trade-off.** We now consider a more classic example, showing how a DM could use COPA to choose a trade-off between accuracy and fairness in classification problems, two objectives defined in completely different ways (Zafar et al., 2017). To this end, we reproduce the experiment from Maheshwari

& Perrot (2022) using FairGrad, whose hyperparameter $\epsilon$ upper-bounds the classifier unfairness, on CelebA (Liu et al., 2015), and create a population of models by sweeping through values of $\epsilon$ and five random seeds.

Fig. 5 (left) shows the front in the objective space using COPA with $p = \infty$, as we navigate it by changing the fairness importance, $\alpha$, showing the trade-off between both objectives. Similar to §5.1, directly solving Eq. 3 always find the solution with maximum accuracy (see Fig. B.5). Instead, COPA helps us uniformly navigate the Pareto front where, e.g., the robust min-max solution ($\alpha = 1/2$) lies in the middle of the front. Moreover, COPA offers a more reliable interpretation of its parameters than the upper-bound given by $\epsilon$, which is clear by observing that, e.g., both $\epsilon = 1$ or $0.25$ yield relatively similar solutions.

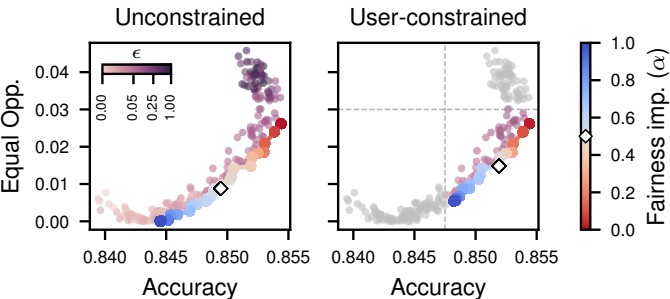

In addition, we consider a more realistic scenario where DMs bargain on acceptable values for the objectives, e.g., a regulatory body could demand equal opportunity to never exceed 0.02 (MacCarthy, 2017). Despite constraining the Pareto front to consider only valid solutions,[2] COPA keeps providing a sensible way to navigate the space of valid models, proving that *we can easily adapt COPA to combine rules on the original and CDF-transformed objective spaces.*

Figure 5: **COPA can be used to meaningfully explore accuracy-fairness trade-offs** in the CelebA experiment from Maheshwari & Perrot (2022) in unconstrained (left) as well as user-constrained scenarios (right).

### 5.3 Case 2: Comparative model analysis

Thus far, we have explored how DMs can meaningfully explore the Pareto front in the context of model selection. Now, we focus on a related but different question: *How much could semantic incomparability change the conclusions drawn from comparative analyses in ML research?*

**1. Incomparable objectives.** First, we consider a MTL setting, where the heterogeneity of the tasks to solve makes it prone to semantic incomparability. In fact, it is common to evaluate models using their average relative performance, $\Delta$, as discussed in §4. To clearly show the issue, we take the multi-SVHN experiment from Javaloy & Valera (2022), based on a modified version of SVHN (Netzer et al., 2011) with a digit on each side of the image, and where we solve three classification tasks: **i)** left digit; **ii)** right digit; and **iii)** parity of their product; and two regression tasks: **iv)** sum of digits; and **v)** density of active pixels in the image.

Fig. 6 shows the ranking of the 14 MTL methods considered by Javaloy & Valera (2022) when we rank them using different criterion functions, namely: COPA with different values of $p$ and equal weights, average relative performance, $\Delta$, and regression error over the density task. The first two columns in Fig. 6 clearly show that the density task dominates the value of $\Delta$, as both rank all methods exactly equal. Similar to the case in Fig. 2, this can be explained as the result of the reference method having nearly zero regression error on the density task, greatly magnifying its relative performance, $\Delta_k$.

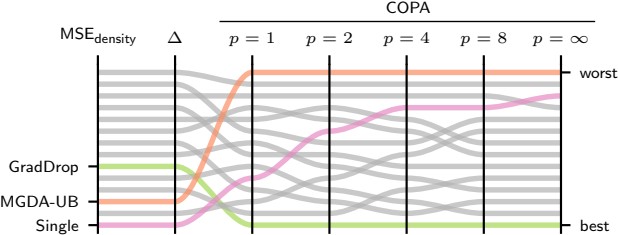

Figure 6: Ranking of MTL methods using different criteria to evaluate them. Methods whose rankings drastically change with $\Delta$ are highlighted in color.

Furthermore, the outlined issue has a significant impact on the conclusions drawn, e.g.: **i)** the *worst* method for all COPA instances, `MGDA-UB` (Sener & Koltun, 2018), becomes the 3rd best method w.r.t. $\Delta$; or **ii)** the best one for every COPA instance, `GradDrop` (Chen et al., 2020), becomes the 6th best. Fig. 6 also shows that the reference method (`Single`) is among the least robust models ($p = \infty$), and slowly improves as we look less at individual performances ($p = 1$). Here, it is worth pointing that the authors were aware of the issue and left the density task out when computing objectives, reporting $\Delta$ and $\text{MSE}_{\text{density}}$ separately.

---

[2]We still use invalid solutions to approximate the CDF of the valid ones.

**2. Seemingly comparable objectives.** Sometimes, semantic incomparability can arise in unexpected scenarios. We take domain generalization as an example and, in particular, the DomainBed (Gulrajani & Lopez-Paz, 2021) experiment from Hemati et al. (2023). Here, the authors compare different methods by training them on some domains, testing them on 4 unseen ones, and reporting the average test domain accuracy, as typically done in the domain generalization literature.

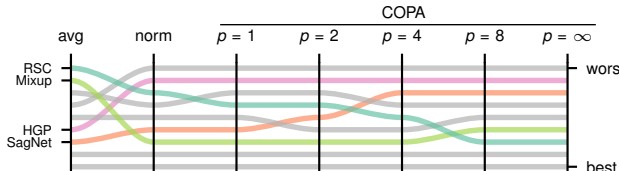

Fig. 7 shows the ranking of different methods as we change the criterion function, with the average accuracy in the first column. For two of the highlighted methods, `RSC` (Huang et al., 2020) and `SagNet` (Nam et al., 2020), we observe their performance deteriorate and improve, respectively, as we consider less robust criteria, being in accordance with the average accuracy for small values of $p$. However, we see a different story with `HGP` (Hemati et al., 2023) and `Mixup` (Wang et al., 2020), whose rankings are consistent for all COPA instances, but drastically change when we average accuracies. Therefore, average accuracy does lead to significantly different analyses concluding, e.g., that `Mixup` is *worse* than `SagNet` and `HGP`, *in disagreement with every other criterion function.*

Figure 7: Ranking of domain generalization methods as we change the criterion function. Average accuracy is inconsistent with every COPA instance.

We can explain this particular case by noting that accuracies present *significantly different ranges* across test domains (see Tab. B.3), and hence differences in domains with less variance become less important when computing the average accuracy. In this case, if we instead normalize the results using $\text{norm}_k$ (Eq. 6), we find in the second column of Fig. 7 that now `Mixup` significantly outperforms `HGP` in these domains on average, swapping their relative rankings and better aligning with all COPA instances. Similar observation have been made in the context of binary classification, where precision depends on the dataset class ratio and thus needs calibration to compare classifiers across datasets (Siblini et al., 2020; Williams, 2021).

### 5.4 Case 3: Benchmarking

Finally, we motivate the use of COPA and CDF-normalized objectives in general in benchmarking settings where, in contrast with the previous use cases, objectives are not necessarily aggregated into a scalar value, but presented together to the user in a plot. Additional figures can be found in §B.5.

We take the AutoML Benchmark (AMLB) as a nice representative which "follows best practices and avoids common mistakes when comparing frameworks" (Gijsbers et al., 2024), and reproduce all figures from the original work, comparing 15 AutoML methods evaluated on 104 objectives. To address incomparability, the authors scaled them using $\text{norm}_k$ (Eq. 6) with a random forest as reference, providing then a number of analyses from these objectives. Remarkably, the authors encourage the use of CD diagrams and Friedman tests, two methods based on relative rankings.

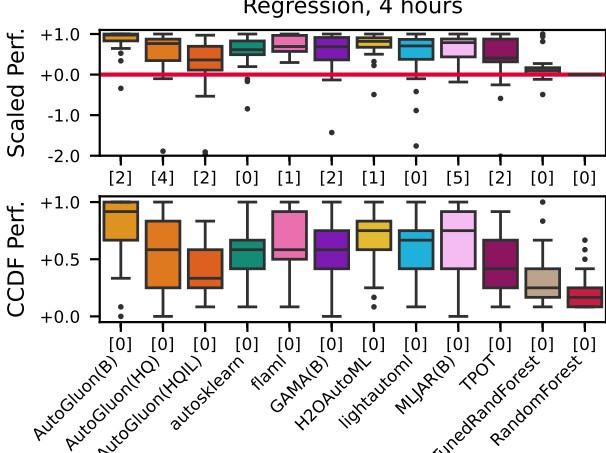

Motivated by the above, a natural step is therefore to use CDF-normalized objectives. Fig. 8 shows the same AMLB boxplot using scaled ($\text{norm}_k$) and CCDF ($1 - F_k(y_k)$) transformed objectives. We observe that CCDFs produce similar plots and come with several benefits: **i)** there are no outliers to report, unlike in the original plot; **ii)** there is no need to pick an arbitrary reference model; and **iii)** it provides clear population-based interpretations, e.g., "on average, `AutoGluon(B)` (Erickson et al., 2020) yields over top-10 % performance

Figure 8: Comparison of AutoML methods on AMLB (Gijsbers et al., 2024) using scaled performance, `norm`, with a random forest as reference method (red line); and using a CCDF-transformation (bottom). Brackets indicate the number of off-view outliers.

on the considered objectives." These benefits extend to all AMLB plots, demonstrating that the proposed CDF transformation is a sensible way of normalizing objectives in general.

# 6 Concluding remarks

In this work, we have shown the importance of having tools that helps us meaningfully navigate the Pareto front in multi-objective ML evaluation, allowing users to perform better-informed decisions, and expanded on which use cases can be interpreted this way. To this end, we have highlighted how crucial is to properly normalize all objectives, and to have a simple criterion function sensibly mapping DM preferences to the Pareto front. We have materialized these insights in the proposed COPA, and extensively demonstrated the potential impact that its adoption can have in areas as fundamental and timely as model selection, comparative model analysis and model benchmarking.

**Limitations.** It is important to note that the presented results rely on a number of assumptions, and their violation can compromise the performance of COPA, e.g., the continuity of the objectives or the i.i.d. assumption for the samples. Also, while the rank-transform works well in our setting, it has been shown not to do so when e.g., modeling variable interactions in statistics (Thompson, 1991). Therefore, the user should carefully consider these nuances before employing COPA and, e.g., if an objective exhibits several modalities and only one of them is acceptable, restrict the use of COPA to consider models within that modality. Moreover, the simplicity of COPA (namely, its criterion function, Eq. 11) might not be appropriate in some scenarios, requiring DM interaction. For example, in §B.1.1 we show that if two objectives are perfectly correlated, COPA can be biased towards them since the CDF-transformation works with marginal information. To overcome this issue, the user should preemptively remove one of the objectives or accordingly tune their weights. Finally, it is important to stress that relative rankings disregard quantitative changes in objectives and, e.g., cannot detect phase changes in objective values. Therefore, and as we demonstrate throughout most of our empirical use-cases, the user should never disregard the original objectives and thoroughly check them before making a final decision to ensure that the selected model meets their expectations.

**Future work.** Our work opens many intriguing venues for future research, e.g., we would be excited to see COPA adapted to active settings with humans-in-the-loop, criterion functions that parametrize more complex preferences, a formal systematization of model selection enabled by COPA, or its adoption in public portals such as the Open LLM Leadearboard (Fourrier et al., 2024) or the DecodingTrust benchmark (Wang et al., 2023). Moreover, future work could expand COPA by using pairwise interactions to normalize the objectives (rather than marginals), or look at other notions in MOO such as proper Pareto optimality (Miettinen, 1999).

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

# Appendix

## Table of Contents

## A  Extended related work and existing approaches

We review the most relevant methods from different research areas, and discuss their similarities and differences with respect to COPA. It is worth stressing that none of the proposed methods, unless otherwise stated, have been applied in the context of multi-objective model evaluation in ML.

**Normalization functions.**  As mentioned in §2, the MOO literature acknowledges that usual optimization objectives are sensitive to the magnitude of their objective values (Miettinen, 1999), and thus they typically addressed it by applying a *component-wise transformation* to normalize their objectives. Two notorious normalization functions are the following:

$$\Delta_k(\mathrm{y}_k) := \frac{\mathrm{y}_k - \mathrm{y}_k^{\mathrm{ideal}}}{\mathrm{y}_k^{\mathrm{ideal}}}, \quad \mathrm{norm}_k(\mathrm{y}_k) := \frac{\mathrm{y}_k - \mathrm{y}_k^{\mathrm{ideal}}}{\mathrm{y}_k^{\mathrm{nadir}} - \mathrm{y}_k^{\mathrm{ideal}}}, \tag{12}$$

where $\mathrm{y}_k^{\mathrm{ideal}}$ is usually approximated and where $\mathrm{y}_k^{\mathrm{nadir}}$ cannot be used for unbounded objectives. Other normalization functions also exist and, in the context of multi-criteria decision making, "max normalization" is also employed (see, e.g., (Wang & Rangaiah, 2025)). Assuming—as it is usual in their literature—to have a finite set of observations and a *maximization* MOO problem (rather than minimization, as in this work), max normalization is defined as:

$$\mathrm{max}_k(\mathrm{y}_k) = \begin{cases} \mathrm{y}_k/\mathrm{max}_h\,\mathrm{y}_k & \text{if } k\text{-th objective needs to be maximized,} \\ \mathrm{min}_h\,\mathrm{y}_k/\mathrm{y}_k & \text{if } k\text{-th objective needs to be minimized.} \end{cases} \tag{13}$$

In our experiments, we noticed that methods using max normalization did not work well. To obtain the results displayed in Fig. 1, we adapted the normalization function to a *minimization* MOO problem as:

$$\mathrm{max}_k(\mathrm{y}_k) = \begin{cases} \mathrm{max}_h\,\mathrm{y}_k/\mathrm{y}_k & \text{if } k\text{-th objective needs to be maximized,} \\ \mathrm{y}_k/\mathrm{min}_h\,\mathrm{y}_k & \text{if } k\text{-th objective needs to be minimized.} \end{cases} \tag{14}$$

**Multi-criteria decision making.**  Continuing with multi-criteria decision making, we compared COPA in Fig. 1 with the following two baselines described in (Wang & Rangaiah, 2025, Chapter 8). First, simple additive weighting (SAW) (MacCrimmon, 1968), which is defined as the weighted average of max-normalized objectives:

$$\mathrm{SAW}(\boldsymbol{\omega}) := \min_{i \in \{1,2,\ldots,N\}} \sum_{k=1}^{K} \omega_k \mathrm{max}_k(\mathrm{y}_{k,i}). \tag{15}$$

Second, analytic hierarchy process (AHP) (Saaty, 1990; 1977) which, intuitively, performs the same max-normalized weighted sum as SAW, but in-between those two steps $\max_k(y_k)$ is substituted by another "score" using a spectral criterion. Specifically, AHP is described by the following algorithm:

1. Max-normalize each objective, $z_k := \max_k(y_k)$.

2. For each objective with index $k$:

   (a) Compute a pairwise comparison matrix for the $k$-th objective $\boldsymbol{A} \in \mathbb{R}^{N \times N}$ as follows:

      i. If $z_{k,i} \geq z_{k,j}$ then
      $$A_{i,j} := \frac{\ln(z_{k,i}) - \ln(z_{k,j})}{\ln(\max_l z_{k,l}) - \ln(\min_l z_{k,l})} \cdot (9-1) + 1 \,.$$

      ii. Otherwise, $A_{i,j} := 1/A_{j,i}$.

   (b) Compute the principal direction of $\boldsymbol{A}$, $\boldsymbol{v}'_k$, such that $\boldsymbol{A}\boldsymbol{v}'_k = \lambda_{\max}\boldsymbol{v}'_k$.

   (c) Normalize $\boldsymbol{v}'_k$ to add up to one, $\boldsymbol{v}_k := \boldsymbol{v}'_k / \sum_i \boldsymbol{v}'_{k,i}$.

3. Return the element that minimizes the weighted sum of scores:

$$\mathrm{AHP}(\boldsymbol{\omega}) := \min_{i \in \{1,2,\ldots,N\}} \sum_{k=1}^{K} \omega_k \boldsymbol{v}_{k,i} \,. \tag{16}$$

It is worth noting that we have adapted the last step to use a predetermined weight vector $\boldsymbol{\omega}$. In the original formulation (Saaty, 1990; 1977), the weight vector is derived similar as how the scores $\boldsymbol{v}_k$ were computed, using a given objective comparison matrix where the DM makes pairwise comparisons across objectives describing which one is more important to them.

We did not compare with other methods in the multi-criterion decision making literature due to their similarity with other methods. For example, the multiplicative exponent weighting (MEW) method (Miller & Starr, 1969) consists on computing the geometric mean (rather than the arithmetic one) of max-normalized objectives. Another example is the "faire un choix adéquat" (FUCA) method (Fernando et al., 2011), which corresponds to COPA with $p = 1$.

**Rank-based approaches.** Following the previous paragraphs, there are other methods than FUCA in different fields that are based on rankings. For example, Yamada et al. (2024) proposed in the context of evolutionary algorithms two different utility functions. The first one corresponds to COPA with $p = \infty$ and only using Pareto-optimal points for the rank transformation, as explained in §4. The second one corresponds to a usual weighted 2-norm using rank-transformed objectives, which we compare with in Fig. B.3 for $p = 8$. Similarly, in the context of multi-objective optimization (MOO), Kukkonen & Lampinen (2007) proposed to use rank-transformed objectives with $p = 1$ (i.e., COPA with $p = 1$) or with $p = -\infty$.

**Ad-hoc normalization functions.** Finally, it is worth recalling some of the ways researchers in the past have come up with ways of normalizing objectives for their specific use cases. Caruana & Niculescu-Mizil (2004) proposed in the context of multi-objectives supervised learning a "general-purpose metric" combining squared error, accuracy, and ROC area (therefore denoted as SAR), which is defined as the arithmetic metric of the three quantities. §§2 and 5 already discussed at length why combining metrics in this way might not be the best idea in general. In the context of evaluating generative models in tabular data, where each column has a different data type, Nazabal et al. (2020) proposed to use the arithmetic mean of the following errors objectives, depending on the data type:

1. Normalized root mean squared error (NRMSE) for numerical variables, defined as:
$$\frac{\sqrt{1/N \sum_i (\mathrm{x}_{k,i} - \hat{\mathrm{x}}_{k,i})^2}}{\max_j \mathrm{x}_{k,j} - \min_j \mathrm{x}_{k,j}} \,, \tag{17}$$

   where $\hat{\mathrm{x}}_{k,i}$ represents the model prediction for the $i$-th sample and $k$-th column.

2. Accuracy for the categorical variables.

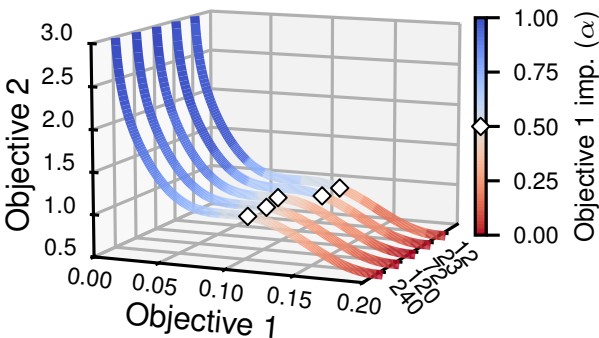 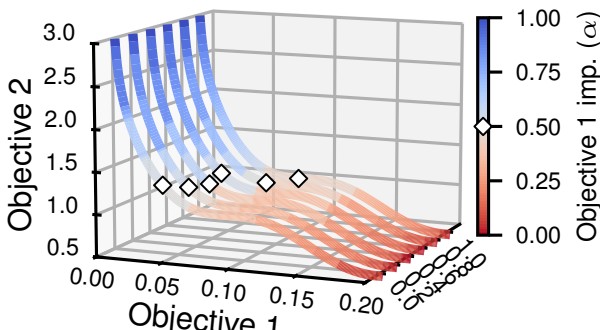

Figure B.1: Synthetic experiment showing the Pareto front using COPA with $p = \infty$ as we change the number of sampled points. While it can be observed a deterioration on the estimated Pareto front (see quantized colors as we reduce $N$), COPA offers a robust estimator even with 12 datapoints.

Figure B.2: Synthetic experiment showing the Pareto front produced by COPA with $p = 2$ as we add an extra objective and change its correlation with Objective 2. Since COPA does not model interactions between objectives, high correlations can skew the retrieved Pareto front if not accounted for through the weights.

3. Displacement for the ordinal variables, defined as the mean absolute error normalized by the range of values that each column can take.

When comparing binary classifiers based on their performance in multiple datasets, it has been shown (Siblini et al., 2020; Williams, 2021) that their precision depends on the ratio of positive and negative samples present in the test set, and therefore it has been proposed to normalize their precision by first re-scaling them, thus enabling the proper comparison of these classifiers across datasets.

In the context of LLM evaluation, it is interesting to revisit the way that the OpenLLM Leaderboard (Fourrier et al., 2024)—which we used for our use cases in §§1 and 5—normalizes the values of their objectives to perform the score average reported in their website. The following is a summary of the process described in their website:

- If the task (i.e., criterion) does not subtasks (e.g., GPQA or MMLU-PRO), apply $\text{norm}_k$ at the beginning of this section, estimating the nadir point.

- If the task has subtasks (e.g., MUSR or BBH), normalize each subtask first, and then average their scores.

- Generative tasks require a different approach. The MATH task uses exact match accuracy, and IFEval uses strict accuracy.

As a result, the average score reported in the leaderboard (alongside the $CO_2$ cost) is the average of $\text{norm}_k$-normalized objectives, with some of them being also the average of $\text{norm}_k$-normalized sub-objectives. Similarly, DecodingTrust (Wang et al., 2023), another benchmark for LLM evaluation, normalizes each of its eight objectives in a different way to make them comparable, as we describe in detail in §B.6.

# B  Experimental details and additional results

In this section, we provide all details to reproduce the experiments presented in the manuscript, as well as additional results which were omitted from the main paper due to space constraints.

## B.1  Synthetic evaluation

As we describe in the main text, for the synthetic experiment we consider the following parametric curve:

$$y_2 = 0.25 \cos(39 y_1^{0.85}) - \log(y_1) - 0.46 \,, \tag{18}$$

where $y_1 \sim \mathcal{U}(0.02, 0.2)$. As a result, we end up with a non-convex Pareto front with a flat area around $y_1 = 0.1$, and two objectives with significantly different distributions. Moreover, the distribution of both objectives are significantly different. Specifically, the first objective is uniformly distributed, while the second one is precisely the plotted curve (if we flipped it to have the second objective as the x-axis), therefore being heavy tailed with

most density lying in the $[0, 0.2]$ interval. The uneven and long-stretch of the domain of the second objective explains why, despite applying $\text{norm}_k$, we still get a biased optimization problem in Fig. 2, as discussed in §2.

### B.1.1 Additional results

**Robustness to sample size.** Despite having a closed-form expression for the variance of our estimator $\text{u}_k$ in §3.2, we empirically show in Fig. B.1 the estimated Pareto front using COPA with $p = \infty$ as a function of the first-objective importance, $\alpha$, as we change the total number of points sampled to estimate it, $N$. We can observe that, despite considerably reducing the number of samples from 240 to 12 datapoints, the estimate given by COPA remains perfectly consistent.

**Robustness to objectives correlation.** Since the normalization function of COPA uses only marginal information, correlations between tasks are only taken into account by the criterion function in Eq. 11. To see the effect that unaccounted correlations could have in the user decisions, we introduce an additional task to our synthetic setting. The data os this new task is generated from a random standard gaussian conditioned on the values of Objective 2 and a correlation coefficient $0 \le \rho \le 1$, which we vary in increments of 0.2. Fig. B.2 shows the plot of one random run using COPA with $p = 2$ and $\alpha = 0.5$, while Tab. B.1 shows the results averaged over 100 randomly sampled third objective.

We observe that, as we increase the correlation between Objective 2 and 3, the performance on Objective 1 worsens despite us not changing the value of $\alpha$. Essentially, the correlation between both objectives skews the models obtained by the simple criterion function we propose in Eq. 11. This effect is more significant as we reduce the value of $p$ (recall the intuition that lower values of $p$ look more at the general objective performance, §3.3), and we did not notice at all with $p = \infty$ (the robust min-max version), but this may not be the case if, e.g., we add more correlated objectives.

This experiment therefore reinforces the idea discussed in §6 that we should always look the original objective values as a last step before deciding whether that is the model we are satisfied with. Moreover, this limitation also makes it clear that future work should look at criterion functions that are aware of the dependencies between objectives.

Table B.1: Retrieved objectives of the synthetic experiment with $p = 2$ and $\alpha = 0.5$ (white marker in Fig. B.2) as we add a third dimension correlated with Objective 2 by $\rho$. Results show mean and standard deviation after repeating the experiment 100.000 times (i.e., resampling the correlated new dimension).

| Correlation $\rho$ | 0.0 | 0.2 | 0.4 | 0.6 | 0.8 | 1.0 |
|---|---|---|---|---|---|---|
| Objective 1 | $0.041 \pm 0.004$ | $0.044 \pm 0.005$ | $0.047 \pm 0.006$ | $0.051 \pm 0.008$ | $0.057 \pm 0.008$ | $0.082 \pm 0.000$ |
| Objective 2 | $1.255 \pm 0.075$ | $1.222 \pm 0.080$ | $1.194 \pm 0.082$ | $1.149 \pm 0.097$ | $1.100 \pm 0.063$ | $1.017 \pm 0.000$ |
| CDF Obj. 1 | $0.204 \pm 0.022$ | $0.205 \pm 0.031$ | $0.215 \pm 0.031$ | $0.239 \pm 0.041$ | $0.286 \pm 0.042$ | $0.402 \pm 0.000$ |
| CDF Obj. 2 | $0.796 \pm 0.022$ | $0.795 \pm 0.031$ | $0.785 \pm 0.031$ | $0.761 \pm 0.041$ | $0.712 \pm 0.047$ | $0.536 \pm 0.000$ |

### B.2 Open LLM Leaderboard: Navigating the LLM performance-cost Pareto front

**Dataset details.** In order to conduct our experiments, we retrieved the publicly available results from the Open LLM Leaderboard (Fourrier et al., 2024) using Huggingface's dataset Python package and, for reproducibility purposes, saved a local copy with the state as of the 9th of January 2025. From the 2929 total LLMs, we discard those which were not publicly available on Huggingface's hub. This leave us with a total of 2148 models, which we use to conduct the experiments described in this work.

**Experimental details.** As explained in the main text, we consider all reported values as objectives. Namely, we take as objectives the $CO_2$ emissions and all 6 benchmark performance scores computed on the following datasets: IFEval (Zhou et al., 2023), BBH (Suzgun et al., 2023), MATH (Hendrycks et al., 2021), GPQA (Rein et al., 2023), MuSR (Sprague et al., 2024), and MMLU-Pro (Wang et al., 2024). Then, we use COPA with $p = \infty$ to produce both Figs. 1 and 4, setting the values of $\boldsymbol{\omega}$ according to the importance given to $CO_2$ emissions, $\alpha$, as $\boldsymbol{\omega} := [\alpha, \frac{1-\alpha}{6}, \ldots, \frac{1-\alpha}{6}]$. To create these figures, we take $10\,000$ values of $\alpha$ evenly-spaced in the unit interval and, since different values of $\alpha$ can provide us with the same model, use their range-average (Fig. 1) or maximum (Fig. 4) as the value to colour the selected LLMs in the figures. There are two

more details worth-discussing. First, in Fig. 1 we use COPA over two objectives (the average score and $CO_2$ emissions) just so that the models selected by all criterion functions lied exactly in the plotted Pareto front, since Pareto-optimal models selected with all $K = 7$ objectives may not be Pareto-optimal when considering this bidimensional representation. Second, we use as y-axis for Fig. 4 the CDF of the $p$-norm computed using the CDF-transformed performance criteria (i.e., of the vector used with COPA, excluding the $CO_2$ dimension), since this represents much more closely the CDF-space that COPA navigates.

### B.2.1 Additional results

**Retrieved models.** Complementing Fig. 4, we present here the quantitative results of those LLMs selected with COPA. In the table we report the reported benchmark scores, a summary of their benchmark performance and $CO_2$, the CDF values found by COPA (same as in Fig. 4), and the value of $\alpha$ used to select these models. As it can be observed, COPA allows us to meaningfully navigate the performance-cost trade-off in the LLM space. Answering the initial question we posed in §1, if we were a practitioner trying to select a balanced LLM in terms of its performance and cost without further prior expectations, *we* would proceed in this case by using COPA with $p = \infty$ and $\alpha = 0.5$, which would yield us a model, unsloth/Phi-3-mini-4k-instruct, in the top-9 % of LLMs in terms of benchmark performance, and top-8 % in terms of $CO_2$ emissions.

Table B.2: Quantitative results of the LLMs highlighted in Fig. 4 from the Open LLM Leadearboard (Fourrier et al., 2024) using COPA with $p = \infty$, as we change the importance of $CO_2$ consumption. Rather than using the average, the CDF value for the performance computes the weighted $\infty$-norm of the CDF-transformed benchmark results (i.e., the value used with COPA but separating $CO_2$ from the rest of objectives).

| | Benchmarks scores | | | | | | Summary | | CDF values | | |
|---|---|---|---|---|---|---|---|---|---|---|---|
| Full model name | IFEval (%) | BBH (%) | MATH (%) | GPQA (%) | MUSR (%) | MMLU-PRO (%) | Average (%) | $CO_2$ cost (kg) | Perf. ($p=\infty$) | $CO_2$ cost | $\alpha$ |
| dfurman/CalmeRys-78B-Orpo-v0.1 | 81.63 | 61.92 | 40.71 | 20.02 | 36.37 | 66.80 | 51.24 | 13.00 | 0.00 | 0.95 | 0.01 |
| maldv/Qwentile2.5-32B-Instruct | 73.93 | 57.21 | 38.07 | 17.90 | 19.96 | 54.21 | 43.55 | 3.53 | 0.01 | 0.87 | 0.02 |
| sometimesanotion/Qwen2.5-14B-Vimarckoso-v3 | 72.57 | 48.58 | 34.44 | 17.34 | 19.39 | 48.26 | 40.10 | 1.93 | 0.01 | 0.79 | 0.03 |
| hotmailuser/FalconSlerp3-7B | 60.96 | 36.83 | 27.42 | 9.17 | 15.90 | 34.75 | 30.84 | 0.61 | 0.05 | 0.19 | 0.21 |
| unsloth/Phi-3-mini-4k-instruct | 54.40 | 36.73 | 15.41 | 9.73 | 13.12 | 33.68 | 27.18 | 0.47 | 0.08 | 0.09 | 0.50 |
| icefog72/Ice0.37-18.11-RP | 49.72 | 31.04 | 6.42 | 8.28 | 12.21 | 23.81 | 21.91 | 0.41 | 0.21 | 0.07 | 0.66 |
| h2oai/h2o-danube3.1-4b-chat | 50.21 | 10.94 | 2.11 | 4.70 | 10.20 | 19.10 | 16.21 | 0.30 | 0.60 | 0.03 | 0.82 |
| postbot/gpt2-medium-emailgen | 14.92 | 3.67 | 0.00 | 1.34 | 6.89 | 1.63 | 4.74 | 0.08 | 0.86 | 0.00 | 0.97 |

**Differences in $p$-norms.** To show the differences between using as criterion function the usual $p$-norm or the one proposed in this paper (Eq. 11), Fig. B.3 shows the same figure as in Fig. 1, comparing the proposed norm in Eq. 11 and the usual weighted $p$-norm with four different normalization functions $\phi$. Note that we do not use $p = \infty$ as in the original figure since, for the usual weighted $p$-norm, $\|\mathbf{u}\|_{\infty,\boldsymbol{\omega}} = \|\mathbf{u}\|_{\infty}$, while for the proposed norm it corresponds to the weighted Tchebycheff problem, as discussed in §3.3.

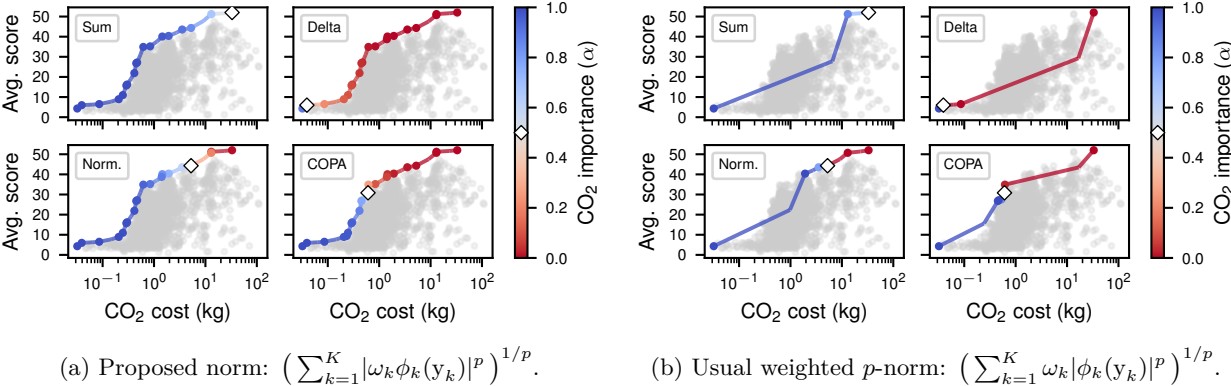

(a) Proposed norm: $\left(\sum_{k=1}^{K} |\omega_k \phi_k(\mathbf{y}_k)|^p\right)^{1/p}$.     (b) Usual weighted $p$-norm: $\left(\sum_{k=1}^{K} \omega_k |\phi_k(\mathbf{y}_k)|^p\right)^{1/p}$.

Figure B.3: LLM experiment comparing the proposed and the usual weighted $p$-norms, with $p = 8$ and taking 10 000 evenly-spaced values for $\alpha$. As discussed after introducing Eq. 11, the usual weighted $p$-norm is not well-suited for our purposes, as CDF-transformed objective lie in the unit interval and quickly vanish.

**Using piece-wise criterion functions.** In the main pages, we exclusively consider criterion functions as weighted norms, as proposed in Eq. 11. To showcase that this is not a real restriction—in fact, we can use any sensible criterion function which we can evaluate—we show in Fig. B.4 the same experiment as in Fig. B.3 where we replaced the criterion function such that it depends on the $CO_2$ footprint of the model we are evaluating. We take this experiment to the extreme, such that the criterion function gives almost-zero importance to $CO_2$ cost if the model consumes less than $0.5\,\text{kg}$, and almost-no-importance to performance otherwise. As we can see, it is still crucial to have semantically comparable objectives, as the rest of normalization functions find a solution far from the decision boundary despite the highly-skewed weights. Furthermore, this experiment shows that we can use any criterion function sensible for the DM.

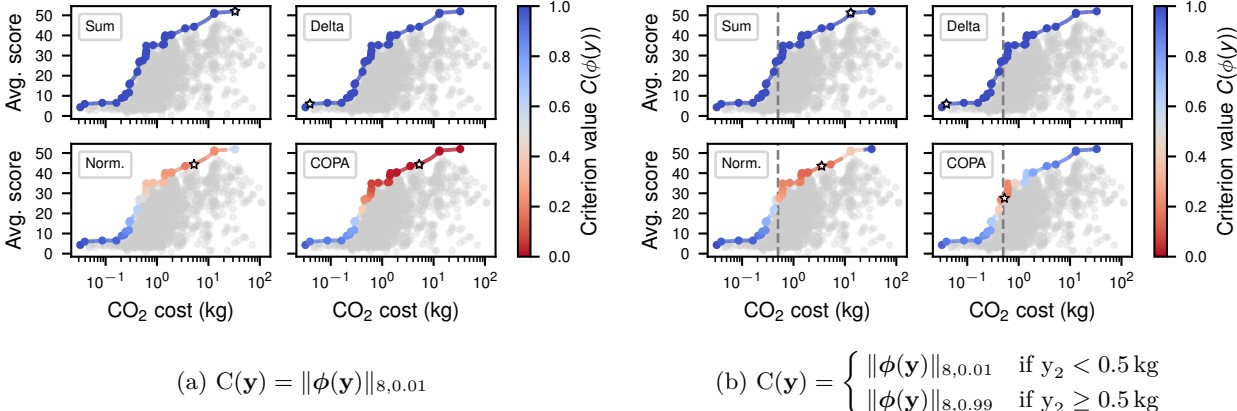

(a) $C(\mathbf{y}) = \|\phi(\mathbf{y})\|_{8, 0.01}$

(b) $C(\mathbf{y}) = \begin{cases} \|\phi(\mathbf{y})\|_{8, 0.01} & \text{if } y_2 < 0.5\,\text{kg} \\ \|\phi(\mathbf{y})\|_{8, 0.99} & \text{if } y_2 \geq 0.5\,\text{kg} \end{cases}$

Figure B.4: LLM experiment where we compare the model retrieved using the proposed weighted $p$-norm, and a piece-wise criterion function that depends on the objectives values. Colors represent the norm value of each model in the Pareto front (i.e., the value of the criterion function), and we look for the point with minimum norm (marked with a star). The importance of having comparable objectives is clear, as only COPA finds a point near the decision boundary despite the highly skewed weights in both norms.

## B.3 Navigating the fairness-accuracy trade-off

**Experimental details.** We reproduce the CelebA (Liu et al., 2015) experiment from (Maheshwari & Perrot, 2022) using their proposed FairGrad algorithm, which code is publicly available at github.com/saist1993/fairgrad, and run this experiment with 10 random initializations and 24 different values of $\epsilon$ (the hyperparameter of FairGrad that represents the desired fairness upper-bound), namely:

$$\epsilon \in \{0.001, 0.002, 0.003, 0.004, 0.005, 0.006, 0.007, 0.008, 0.009,$$
$$0.01, 0.02, 0.03, 0.04, 0.05, 0.06, 0.07, 0.08, 0.09,$$
$$0., 0.1, 0.2, 0.3, 0.5, 1.\}$$

This leave us with a total of 240 models. To produce Fig. 5, we use COPA with $p = \infty$ and 50 values of $\alpha$ evenly-spaced in the unit interval. For the constrained case, we simply drop those points that do not match the requirements for accuracy (being larger than 0.845) and fairness (having an equal opportunity value smaller than 0.02) before selecting any models with COPA. Of course, to compute the rankings of the accuracy, we take into account that it needs to be maximized and used the opposite order relation. Similarly, when we applied other normalization functions (see below), we employ the error rate (rather than the accuracy), so that it has to be minimized.

### B.3.1 Additional results

We show in Fig. B.5 the same plot as in Fig. 5, using all the considered normalization functions and baselines. Similarly to what we observed in the introductory example in Fig. 1, all other methods are biased towards minimizing one of the objectives. It is worth noting, however, than in this case AHP (Saaty, 1990; 1977) and norm do a good job.

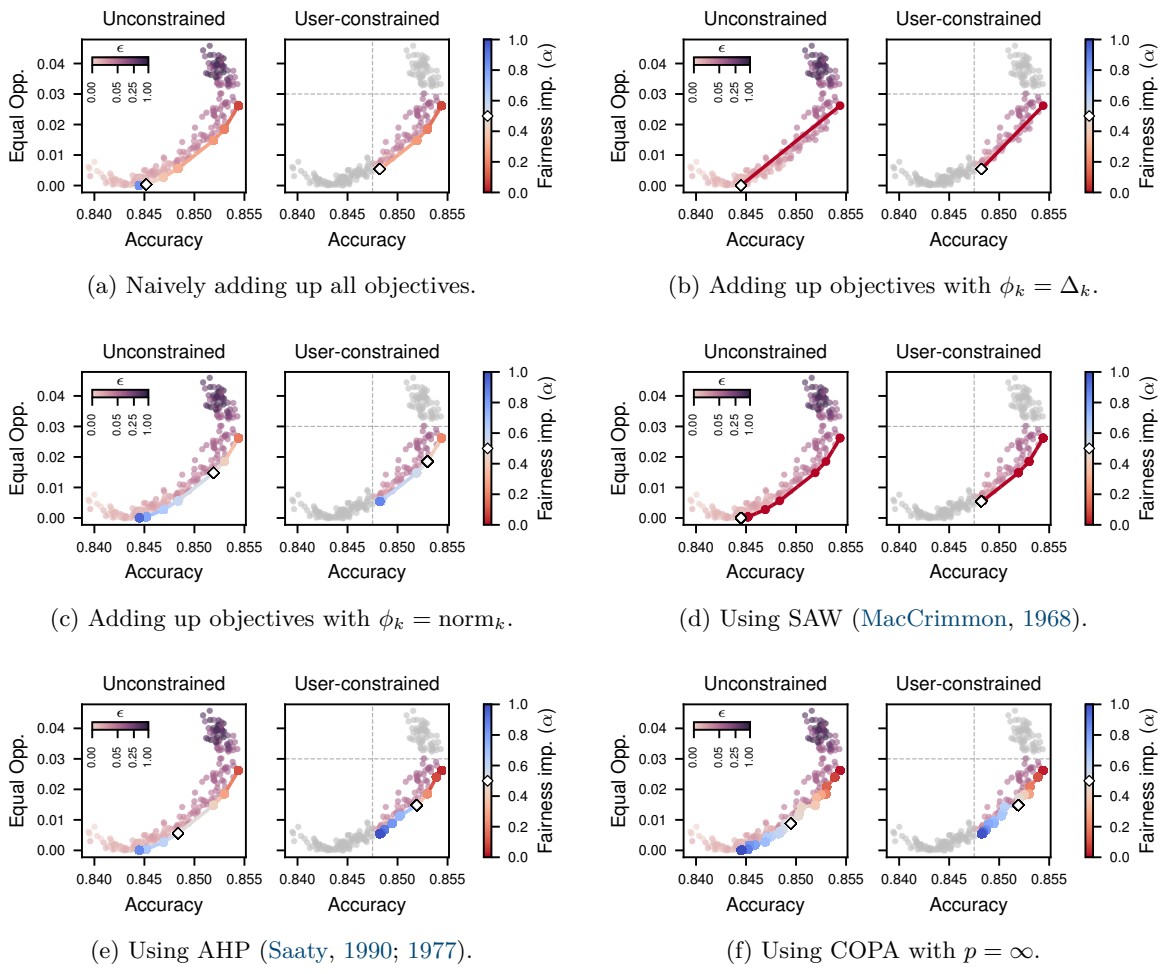

(a) Naively adding up all objectives.

(b) Adding up objectives with $\phi_k = \Delta_k$.

(c) Adding up objectives with $\phi_k = \mathrm{norm}_k$.

(d) Using SAW (MacCrimmon, 1968).

(e) Using AHP (Saaty, 1990; 1977).

(f) Using COPA with $p = \infty$.

Figure B.5: We reproduce the fair ML experiment from §5.2 using different baselines. We can observe that COPA meaningfully navigates the Pareto front, with all other approaches being biased towards one of the extreme solutions to some extent. Indeed, $\Delta_k$ only reaches the two extreme solution despite sampling 50 evenly-spaced values for $\alpha$.

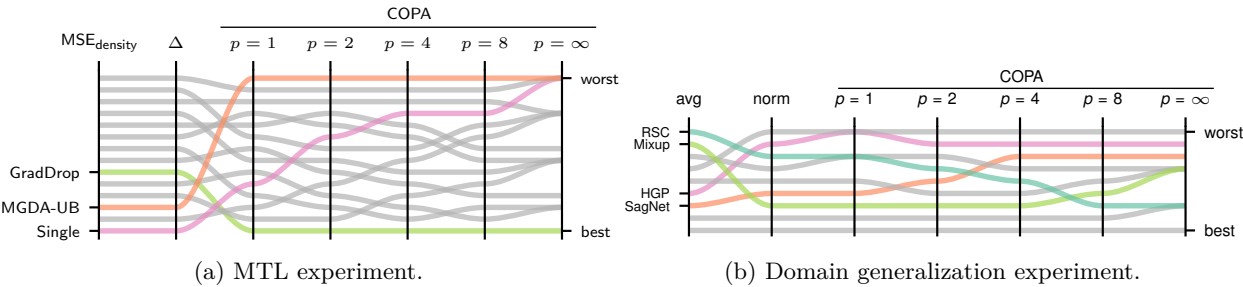

(a) MTL experiment.

(b) Domain generalization experiment.

Figure B.6: Reproductions of Figs. 6 and 7 where we do not untie methods. As it can be observed, conclusions drawn in the main paper do not change and untying only serves aesthetic purposes.

## B.4 Comparative model analysis experiments

**Experimental details.** For the figures shown in §5.3, we retrieved the results reported by two selected works. In particular, we took the values reported in the second half of Table 5 from the work of Javaloy & Valera (2022) for the MTL experiment, and values reported in Table 4 of Hemati et al. (2023) for the domain generalization

Table B.3: Different effective ranges explain the differences in rankings of the domain generalization experiment. The table shows the effective range of each domain accuracy, and the performance of `Mixup` and `HGP` for the raw and normalized (norm$_k$, Eq. 6) domain accuracies, respectively.

|  |  | VLC | PACS | OfficeHome | DomainNet | Avg |
|---|---|---|---|---|---|---|
|  | Min. acc. | 76.30 | 78.80 | 60.20 | 23.40 | - |
|  | Max. acc. | 79.30 | 84.80 | 68.50 | 41.40 | - |
| Acc. | Mixup | 77.70 | 83.20 | 67.00 | 38.50 | 66.60 |
| | HGP | 76.70 | 82.20 | 67.50 | 41.10 | 66.88 |
| Norm. | Mixup | 46.67 | 73.33 | 81.93 | 83.89 | 71.45 |
| | HGP | 13.33 | 56.67 | 87.95 | 98.33 | 64.07 |

experiment of the main text. From these values, we simply re-rank them using the different criterion functions discussed in the main paper, and highlight those which we consider are interesting for the discussion we carry out in the main manuscript. To ease visualization, as ties are more frequent in $p = 1$ and $p = \infty$—especially when we have only a handful of models—we untie by using the ranking of $p = 8$ as a secondary criterion. That is, if two models tie, we rank those by their performance with $p = 8$. We plot the figures without untying in Fig. B.6 for the sake of transparency, showing that conclusions do not change. We use equal weights for all versions of COPA. One important detail is that, for the domain generalization case, we kept only the top methods, as the rest do not add anything more to the discussion and make the plot more difficult to read.

### B.4.1 Additional results

As mentioned just above, we discarded some methods in the domain generalization figure of the main text (i.e., Fig. 7). For completeness, we show in Fig. B.7a the full figure with all methods included, and highlighting `Hutchinson`, the second method proposed by the authors, along `HGP`. Also, we show in Fig. B.7b the same figure but using as data the one reported in Table 9 from Hemati et al. (2023) (instead of Table 4). This table was reported in the supplementary material, and the difference between both tables is the method used to select hyperparameters, with all methods but those proposed by this particular work (i.e., `HGP` and `Hutchinson`) improving their performance. More crucially, we show once again the huge discrepancies in ranking between using the average accuracy and any of the COPA versions. This time, we also report `Hutchinson`, which is the best method for all criterion functions in Fig. B.7a, and the fourth to worst method in Fig. B.7b. We can again observe how much our final conclusions can change in Fig. B.7b, where the fourth to worst method in terms of average domain accuracy, `VREx` (Krueger et al., 2021), is better than `Hutchinson` in all instances of COPA. To finalize, we consider important to report that, in both figures, the first gray line (i.e., the second-best and best methods, respectively) correspond to the domain generalization method named `CORAL` (Sun & Saenko, 2016).

### B.5 AutoML Benchmarking (AMLB) experiment

**Experimental details.** To demonstrate the out-of-the-box utility of COPA and its two components, we reproduce some of the plots from the AutoML Benchmark from Gijsbers et al. (2024). To achieve this, we simply modify the Jupyter notebook publicly available at github.com/PGijsbers/amlb-results, and add a few lines of code to compute COPA as proposed in this work.

### B.5.1 Additional results

To complement Fig. 8 from the main text, we provide here side-by-side comparisons of more figures reported by Gijsbers et al. (2024), further reinforcing the argument of broadly adopting CDF-transformed objectives for general cases.

In particular, we show in Fig. B.8 the same three figures as Figure 3 from the original work, where the same advantages when using the proposed CDF transformation, as those discussed in the main text (see §5.4), can be observed here. Furthermore, we show in Fig. B.9 Figure 4 from the original work, where all 104 objectives are used, further showcasing the benefits of the proposed transformation.

Finally, we also reproduce Figure 7 from the original publication in Fig. B.10, where different Pareto plots are generated according to the type of tasks, showing the performance-speed trade-off, similar in spirit to

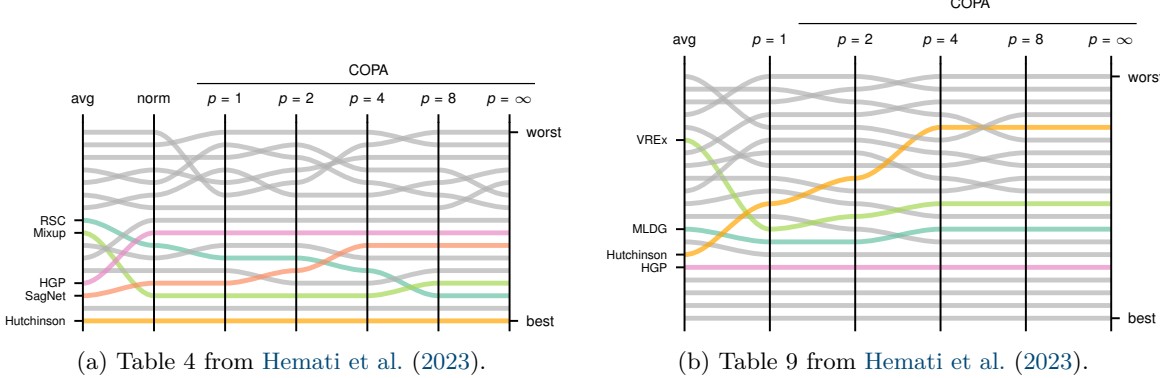

(a) Table 4 from Hemati et al. (2023).  (b) Table 9 from Hemati et al. (2023).

Figure B.7: Ranking of the domain generalization methods considered by Hemati et al. (2023) as we use different criterion functions to rank them. We can appreciate a significant change of rankings, and the average accuracy in particular being highly inconsistent with all versions of COPA. We highlight those methods used for the discussion in the text.

Fig. 1 in this work. Here, we use COPA with $p = 2$ and equal weights. We can observe that, while some of the figures are quite similar, e.g., binary classification in the top row, some others differ significantly, e.g., regression in the bottom row, where COPA reports two less Pareto-optimal models. Beyond the differences in using scaled vs. CDF-transformed objectives, which we have extensively discussed during this paper, and showed the significant advantages of employing the latter, the differences in the number of Pareto-optimal models is due to the fact that the Pareto front is computed *after* aggregating the performance metrics. This is in stark contrast with the approach taken in this work (except for Fig. 1 for visualization purposes, see §B.2), where we compute Pareto-optimal points on the space of all objectives. As we have been arguing during this work, COPA allows us to meaningfully navigate the Pareto front, enabling the creation of plots such as those reported in this work (e.g., Figs. 1, 4 and 5), which are significantly more informative than those reported before our work, as it can be clearly observed in Fig. B.10.

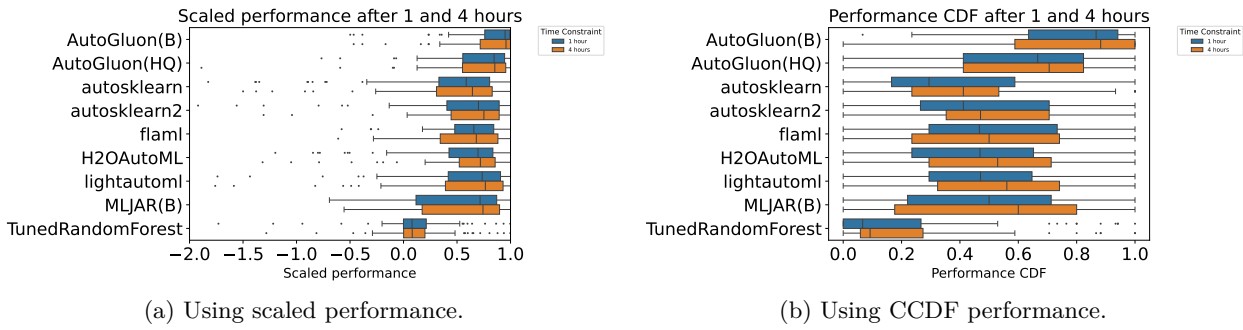

(a) Using scaled performance.

(b) Using CCDF values.

Figure B.8: We reproduce Fig. 3 from Gijsbers et al. (2024) in (a) using their proposed scaled performance, and we show the same figure in (b) but using complementary CDF values (CCDF, one minus the CDF value). The same advantages as those discussed in §5.4 can be observed here.

(a) Using scaled performance.

(b) Using CCDF performance.

Figure B.9: We reproduce Fig. 4 from Gijsbers et al. (2024) in (a) using their proposed scaled performance, and we show the same figure in (b) but using complementary CDF values (CCDF, one minus the CDF value). The same advantages as those discussed in §5.4 can be observed here.

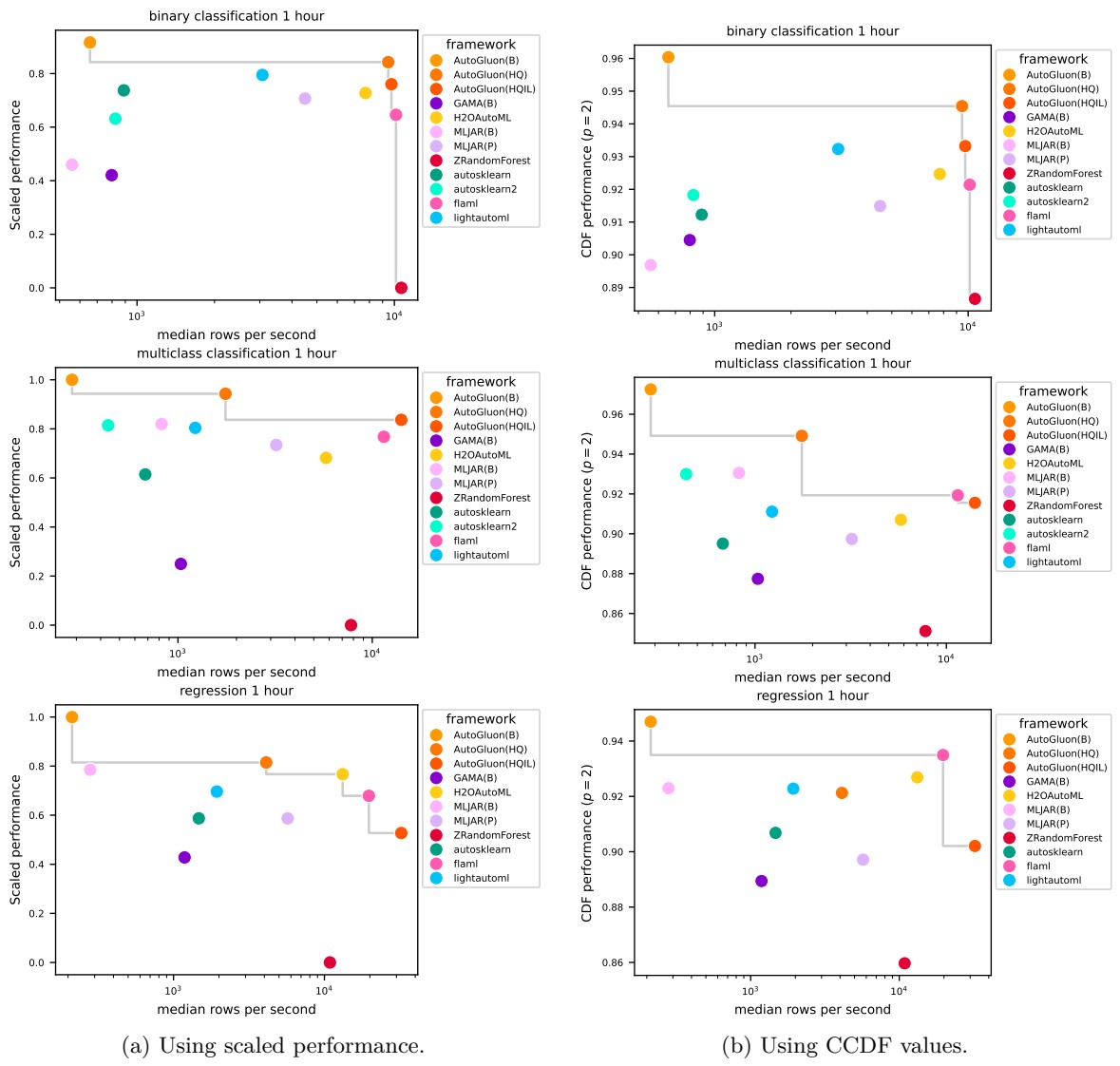

(a) Using scaled performance.

(b) Using CCDF values.

Figure B.10: We reproduce Fig. 7 from Gijsbers et al. (2024) in (a) using their proposed scaled performance, and we show the same figure in (b) but using complementary CDF values (CCDF, one minus the CDF value).

### B.6 DecodingTrust: Navigating the LLM trustworthiness Pareto front

**Dataset details.** We look once more into the LLM space and, this time, we focus on the DecodingTrust leaderboard (Wang et al., 2023). In contrast with the Open LLM leaderboard (Fourrier et al., 2024), DecodingTrust focuses in assessing the trustworthiness of LLM models, rather than on sheer performance. To this end, the authors design a comprehensive list of prompts that the model should successfully answer to. For the interest of this work, it suffices to say that DecodingTrust evaluates LLMs on 8 completely different aspects and the authors had to come up with different ad hoc normalization functions for each of the objectives so that they all lie in the $[0, 100]$ interval. Below there is a summary of the formulas employed by the authors and, while we do not give context for the variables shown in the equations, we want to stress the diversity of scores and normalizations that the authors had to propose to make objectives more comparable. For further details on the normalization functions and, more in general, DecodingTrust, refer to (Wang et al., 2023, App. I):

- *Toxicity*: $1 - \frac{1}{2\sum_i |D_i|} \sum_{i=1}^{4} \sum_{x \in D_i} \left( f(x_{\mathrm{adv}}^*; x) + f(x_{\mathrm{benign}}^*; x) \right)$ .

- *Stereotype bias*:

$$\frac{S_{\mathrm{benign}} + S_{\mathrm{untargeted}} + S_{\mathrm{targeted}}}{3} \quad \text{where} \quad S_{\mathrm{scenario}} = 1 - (\sum_{i=1}^{n_{\mathrm{ST}}} \sum_{j=1}^{n_{\mathrm{DG}}} S_{ij})/(n_{\mathrm{ST}} n_{\mathrm{DG}}) .$$

- *Adversarial robustness*: $\frac{\sum_{i=1}^{T} \mathrm{acc}_i * d_i}{\sum_{i=1}^{T} d_i}$ .

- *Out-of-distribution robustness*: $(\mathrm{ACC}_{\mathrm{style}} + \mathrm{Reliability}_{\mathrm{OOD}} + \mathrm{ACC}_{\mathrm{style}}^{\mathrm{icl}} + \mathrm{ACC}_{\mathrm{domain}}^{\mathrm{icl}})/4$ where:

$$\mathrm{ACC}_{\mathrm{style}} = \frac{1}{S} \sum_{s=1}^{S} \mathrm{acc}_s ,$$

$$\mathrm{Reliability}_{\mathrm{OOD}} = \frac{\mathrm{Reliability}_{2023} + \mathrm{Reliability}_{2023\mathrm{idk}}}{2}$$

$$\mathrm{Reliability}_{\mathrm{setting}} = \mathrm{RR}_{\mathrm{setting}} + (1 - \mathrm{RR}_{\mathrm{setting}}) * \mathrm{macc}_{\mathrm{setting}} ,$$

$$\mathrm{ACC}_{\mathrm{setting}}^{\mathrm{icl}} = \frac{1}{D * N} \sum_{d=1}^{D} \sum_{i=1}^{N} \mathrm{acc}_{di}^{\mathrm{setting}} .$$

- *Robustness to adversarial demonstrations*: $(s^{(\mathrm{cf})} + s^{(\mathrm{sc})} + s^{(\mathrm{bkd})})/3$ where:

$$s^{(\mathrm{cf})} = \frac{1}{|D^{(\mathrm{cf})}|} \sum_{i \in D^{(\mathrm{cf})}} \mathrm{acc}_i^{(\mathrm{Demo+CF})} ,$$

$$s^{(\mathrm{sc})} = \frac{1}{|D^{(\mathrm{sc})}|} \sum_{i \in D^{(\mathrm{sc})}} \frac{\mathrm{acc}_i^{(\mathrm{entail})} + \mathrm{acc}_i^{(\mathrm{non\text{-}entail})}}{2} ,$$

$$s^{(\mathrm{bkd})} = 1 - \frac{1}{|M||B|} \sum_{i \in B} \sum_{j \in M} \mathrm{ASR}_{ij} .$$

- *Privacy*: $1 - (0.4\mathrm{LR}^{(\mathrm{Enron})} + 0.3\mathrm{LR}^{(\mathrm{PII})} + 0.3\mathrm{LR}^{(\mathrm{Understand})})$ where

$$\mathrm{LR}^{(\mathrm{Enron})} = \frac{1}{T} \sum_{t=1}^{T} \frac{\mathrm{LR}_t^{(\mathrm{Email})} + \mathrm{LR}_t^{(\mathrm{Local})} + \mathrm{LR}_t^{(\mathrm{Domain})}}{3} ,$$

$$\mathrm{LR}^{(\mathrm{PII})} = \frac{1}{P} \sum_{p=1}^{P} \overline{\mathrm{LR}}^p ,$$

$$\mathrm{LR}^{(\mathrm{Understand})} = \frac{1}{WE} \sum_{w=1}^{W} \sum_{e=1}^{E} \overline{\mathrm{LR}}_{we} \,.$$

- *Machine ethics*: $(\mathrm{ACC}^{\mathrm{zero}} + \mathrm{ACC}^{\mathrm{few}} + (1 - \overline{\mathrm{FPR}}^{\mathrm{jailbreak}}) + (1 - \overline{\mathrm{FPR}}^{\mathrm{evasive}}))/4 \,.$

- *Fairness*: $100 \left( 1 - \dfrac{M_{\mathrm{dpd}}^{(\mathrm{zero})} + M_{\mathrm{dpd}}^{(\mathrm{few\text{-}unfair})} + M_{\mathrm{dpd}}^{(\mathrm{few\text{-}fair})}}{3} \right) \,.$

**Additional results.** After computing all the variables above, we end up with 8 objectives, which are aggregated in the their official leaderboard by taking the average of all objectives (i.e., using $p = 1$ in Eq. 11 since all values are non-negative). One natural question then is how do the rankings obtained with the average score change if we plug in COPA on the objectives given above. To this end, we take the results published in the official site of DecodingTrust and recompute their rankings using as aggregated scores COPA with $p \in \{1, 2, 4, 8, \infty\}$. While this can be easily done in the full leaderboard with 55 models to this date (and it is indeed done this way in the code accompanying this manuscript), we show in Fig. B.11 a bar plot showing the differences in ranking of a small subset of representative models w.r.t. average ranking. We summarize the main takeaways as follows:

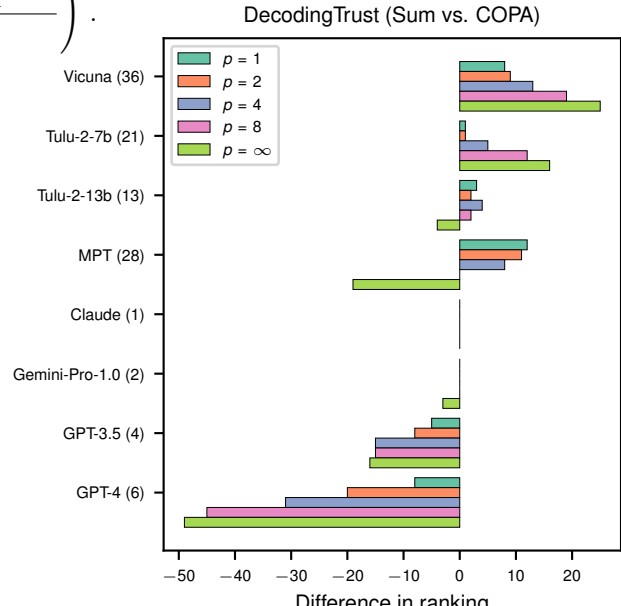

Figure B.11: Difference in ranking for a subset of models on the DecodingTrust leaderboard (Wang et al., 2023) for different $p$ values of COPA, where the number within parenthesis indicate their original ranking.

1. *Differences in normalization*: Even with $p = 1$, we observe significant differences in COPA w.r.t. the original ranking where, e.g., Vicuna and MPT rank around 10 positions better and the two GPT models close to 10 worse.

2. *Robustness with $p$*: As discussed in the manuscript, increasing (resp. decreasing) $p$ can be interpreted as adding (removing) importance to the performance of the models on individual objectives. We observe a similar trend here, where those models that lack in one of the specific objectives (e.g., GPT-4 which performs the worst in the Fairness objective) start losing ranks as we increase $p$, and those which are more robust are rewarded instead (e.g., Vicuna and Tulu-2-7b).

3. *Robustness trend.* Similar to those experiments from §5.3, we observe a clear correlation between the rankings and the values of $p$, that is, the aforementioned robustness criterion is stressed as we increase $p$. While this is not always the case, e.g., Tulu-2-13b fluctuates by one or two rankings as we change $p$, the trend is rather clear.

4. *Quasi-dominant models.* While there is no a dominant model (i.e., one which is better for all objectives), some models like Claude and Gemini-Pro-1.0 rank top-10 for all but, respectively, one and two objectives. As a result, we see that they consistently rank first and second for almost all values of $p$, just like if we took the average. This is, in part, a result of the CDF estimator having less variance for more extreme samples (see Prop. 3.1). In layman's terms, a model that does almost everything great is easy to spot.

5. *Diversity of solutions.* While not shown in Fig. B.11, it is interesting to remark that in this setting, with 8 diverse objectives, we find that out of the 55 available models, the best-worst performing model (that is, the most robust model according to COPA with $p = \infty$) achieves a worst-ranking of 22, with the two second-best LLMs obtaining a worst-ranking of 40. That is, only with 8 objectives we already find that any model performs relatively bad in at least one of them.

