# OpenReview forum: "COPA: Comparing the incomparable in multi-objective model evaluation"
_TMLR — Rejected by TMLR_

### Review · Reviewer_PZuw · 2025-12-08

**Summary Of Contributions:**

The paper makes the following contributions:

1. The authors clearly articulate why existing practices—such as weighted sums, heuristic normalizations, or relative performance scores—can produce biased or unstable rankings when objective scales and distributions differ. They frame this issue as semantic incomparability and show how it affects model selection, comparative analysis, and benchmarking.
2. COPA applies a rank-based CDF transformation to each objective, ensuring that all normalized objectives follow a uniform distribution in [0,1]. This provides an objective-agnostic, strictly monotonic normalization that preserves Pareto optimality while offering a population-based interpretation (e.g., “top-x% performance”).
3. he authors propose a weighted p-norm over normalized objectives, where the weight vector ω encodes objective importance and the parameter p controls robustness. This yields a simple yet expressive mechanism for exploring the Pareto front that generalizes classical criteria such as weighted averages and Tchebycheff norms.
4. Through case studies on LLM selection (performance vs. CO₂), fairness–accuracy trade-offs, multitask learning, domain generalization, and AutoML benchmarking, the paper demonstrates that COPA avoids the pathological behaviors of traditional normalization schemes and often leads to substantially different—and more sensible—comparative conclusions.

**Audience:**

Yes

**Audience Explanation:**

The paper addresses a broadly relevant problem in multi-objective model evaluation and proposes a simple, general, and practically useful framework. Many researchers working on model selection, benchmarking, fairness, domain generalization, AutoML, and LLM evaluation would benefit from understanding how semantic incomparability affects current practices and how COPA provides a principled alternative. The empirical demonstrations across diverse subfields further increase its relevance to a wide portion of TMLR’s audience.

**Broader Impact Concerns:**

The paper presents a methodological contribution for multi-objective model evaluation and does not introduce new datasets, new deployment settings, or models that generate or manipulate user-facing content. As such, it does not appear to raise significant ethical or societal risks beyond those typical of evaluation frameworks. One possible consideration is that improved evaluation tools may influence downstream model selection in high-stakes domains; however, the method itself is agnostic and does not promote any harmful application. Overall, there are no notable broader impact concerns beyond standard considerations for fair and transparent evaluation practices.

**Claims And Evidence:**

Yes

**Claims Explanation:**

The paper provides clear theoretical arguments and supports them with diverse and carefully designed empirical studies. The motivation for semantic incomparability is well illustrated through concrete failure cases of existing normalization and aggregation methods. The proposed CDF-based normalization is grounded in established statistical principles, and its properties are clearly stated. The experimental section is extensive, covering multiple realistic settings—LLM selection, fairness–accuracy trade-offs, multi-task learning, domain generalization, and AutoML benchmarking—where the claimed advantages of COPA are repeatedly observed. These experiments convincingly demonstrate that previous evaluation practices can lead to biased or misleading conclusions, whereas COPA yields more stable and preference-aligned results. While some limitations remain (e.g., reliance on rank-based marginal approximations and assumptions about continuity), the main claims are consistently supported by the evidence presented.

**Requested Changes:**

1. The method relies on objectives being approximately continuous and i.i.d. across models. While the paper acknowledges these assumptions, it would be helpful to more explicitly discuss scenarios where these assumptions may break (e.g., discrete or highly multimodal objectives, strong correlations), and provide guidance or mitigation strategies.
2. Since COPA applies marginal normalization independently, interactions between objectives are not explicitly modeled. A short discussion or experiment illustrating the behavior under high correlation would strengthen the methodological transparency.

---

> ### Author Response · Authors · 2026-03-31
>
> We truly appreciate the reviewer’s feedback. It is great to hear that both our theoretical and empirical arguments are convincing and support the benefits of COPA in assisting the user in mapping their preferences to the Pareto front. We are particularly pleased to know that our claims are repeatedly observed in the experiments we designed over various ML settings.
>
> > It would be helpful to more explicitly discuss scenarios where these assumptions may break (e.g., discrete or highly multimodal objectives, strong correlations), and provide guidance or mitigation strategies.
>
> To address both changes requested by the reviewer, we have extended the limitations paragraph discussing more limitations the user could face when using COPA, as well as actionable items the user could employ to alleviate these issues. The extended paragraph appears in the concluding remarks and we have introduced the following new text (plus other small edits, see the updated version):
>
> Therefore, the user should carefully consider these nuances before employing COPA and, e.g., if an objective exhibits several modalities and only one of them is acceptable, restrict the use of COPA to consider models within that modality. Moreover, the simplicity of COPA (namely, its criterion function, Eq. 11) might not be appropriate in some scenarios, requiring DM interaction. For example, in §B.1.1 we show that if two objectives are perfectly correlated, COPA can be biased towards them since the CDF-transformation works with marginal information. To overcome this issue, the user should preemptively remove one of the objectives or accordingly tune their weights.
>
> > Since COPA applies marginal normalization independently, interactions between objectives are not explicitly modeled. A short discussion or experiment illustrating the behavior under high correlation would strengthen the methodological transparency.
>
> This is a great point. While this already happens to a certain extent in real-world experiments, it is a bit hard to visualize and especially control. We have included an additional experiment on the synthetic use-case where we introduce additional objectives that are perfectly correlated with “Objective 2” and, as we include more of them, the front skews more towards the second objective. We mention this new experiment in the limitation section.

---

### Review · Reviewer_MHhY · 2026-01-12

**Summary Of Contributions:**

Summary

This paper introduce COPA, which addresses heterogeneous multi-objective evaluation by combining rank-based CDF normalization with a weighted $p$ -norm aggregation. The work demonstrates that this approach yields more robust Pareto front exploration and distinct ranking outcomes compared to traditional linear scaling methods across various ML domains.

Strengths
- Addresses a practical and widely encountered problem in multi-objective model evaluation, particularly in settings involving heterogeneous metrics with different units and scales.
-  Simple and easy to implement, leveraging rank-based normalization and standard aggregation.
- Broad empirical validation across multiple domains illustrates the method's generality.

Weaknesses
-  Limited methodological novelty, as the core idea primarily combines rank-based normalization with standard aggregation techniques rather than introducing a fundamentally new evaluation principle.
- Strong reliance on normative assumptions, particularly the use of rank-based semantics, which discard magnitude information and may not be appropriate in many decision-making contexts.
- The experiments largely demonstrate that COPA produces more "uniform" Pareto fronts—a property mathematically guaranteed by its design—but fail to provide independent evidence (e.g., downstream utility or expert agreement) that this uniformity actually translates to superior decision quality compared to baselines.
- The analysis often conflates difference with improvement by implying that because COPA yields different rankings than traditional baselines, the baselines are "biased" and COPA is "correct," yet there is no ground truth provided to substantiate that the new rankings are empirically better.
- The approach is underexplored regarding constraint-driven deployment scenarios, as softening absolute metrics into relative ranks can obscure critical failures where hard thresholds or regulatory requirements (e.g., safety limits) must take precedence over relative trade-offs.

**Audience:**

Yes

**Audience Explanation:**

The paper addresses an issue that is relevant to a segment of the TMLR audience, namely how normalization and aggregation choices can influence conclusions in multi-objective model evaluation and benchmarking. Even if the proposed solution and claims are subject to debate, the empirical examples highlight that seemingly innocuous evaluation design decisions can substantially affect rankings and interpretations. As such, the paper may be of interest to researchers working on evaluation methodology, benchmarking, or model selection in multi-objective settings.

**Broader Impact Concerns:**

No specific broader impact or ethical concerns are identified. The work is primarily methodological and focuses on evaluation and aggregation of existing performance metrics. It does not introduce new models, data, or deployment mechanisms, nor does it appear to raise ethical risks beyond standard considerations associated with model evaluation and benchmarking.

**Claims And Evidence:**

No

**Claims Explanation:**

While the paper provides a wide range of empirical demonstrations illustrating the behavior of COPA under the proposed evaluation framework, many of its central claims are inherently normative and are primarily supported using criteria that are closely aligned with the method’s design. As a result, the evidence is largely illustrative rather than independently convincing.

In particular, the experiments focus on qualitative properties such as smooth preference–front navigation and ranking stability under different aggregation schemes, but do not establish that COPA leads to superior decision quality, better alignment with real decision-maker preferences, or improved downstream utility under independently motivated evaluation criteria.

**Requested Changes:**

- The paper should substantially revise and narrow its claims. Many of the current claims are normative (e.g., regarding improved meaningfulness or interpretability of trade-offs) and are not adequately supported by independent evidence. The authors should clearly delineate which properties are design choices versus empirically validated advantages.

- The evaluation methodology should be strengthened with criteria that are independent of COPA’s design goals. Beyond illustrative re-rankings and visualizations, the paper should provide evidence that COPA leads to improved decision quality, better alignment with external preferences, or more appropriate outcomes under clearly defined evaluation objectives.

- The presentation would benefit from clearer separation between descriptive observations and prescriptive recommendations, particularly in the discussion of experimental results.
-  Additional discussion of when COPA is not appropriate (e.g., constraint-driven or regulatory settings) would help better position the scope of the method.

---

> ### Author Response · Authors · 2026-03-31
>
> We appreciate the reviewer feedback and their acknowledgement on the importance of the problem we tackle. In particular, we are happy to see that the reviewer agrees our work highlights the existing flaws in multi-criteria evaluation and its practical consequences. Next, we address the reviewer concerns:
>
> > Limited methodological novelty, as the core idea primarily combines rank-based normalization with standard aggregation techniques rather than introducing a fundamentally new evaluation principle
>
> We would like to recall that simplicity should not be a reason for rejection. We clearly state our contributions at the end of section 1, among them, highlighting the potential harms that unaddressing incomparability in multi-criteria evaluation can have and clearly showing the extent of this problem through different ML use cases. However, note also that our aggregation function is not the standard weighted $p$-norm, and that we clearly show their differences in Figure B.2.
>
> > Strong reliance on normative assumptions, particularly the use of rank-based semantics, which discard magnitude information and may not be appropriate in many decision-making contexts.
>
> As the reviewer rightly pointed out, discarding magnitude information is not appropriate in many situations. Please note that we use ranks to aggregate values across different objectives, but we do not advocate disregarding this information, as we explicitly acknowledge in the limitations paragraph (section 6). This is why, for example, we use the original objectives in most Pareto plots: In Figure 1 (where we can still plot both objectives), this is where the user can decide whether they are satisfied with the provided model, and nudge the decision (by changing $\alpha$) otherwise. Moreover, we show how to use the original objectives in combination with COPA in the constraint-driven fairness use case (Figures 5 and B.4) as well as in the objective-dependent criterion in Figure B.3 (right).
>
> > The analysis often conflates difference with improvement [..] there is no ground truth provided to substantiate that the new rankings are empirically better.
>
> The problem we address (mapping user preferences to the “expected” model) is hard to assess quantitatively. Instead, we try that our claims are based on clear qualitative results observed repeatedly across experiments, as acknowledged by the rest of reviewers (“The claims are generally supported by empirical results across several application domains”, “The experimental section is extensive, covering multiple realistic settings where the claimed advantages of COPA are repeatedly observed”). For example, when we talk about the induced bias of other methods and check SAW in Figure 1, one can see that to start choosing methods with good performance the user would need to set $\alpha$ to values close to $0.1$, which seems counter-intuitive to us: a value of $\alpha = 0.5$ should already strike a right balance between both objectives, whereas $\alpha = 0.1$ should map to pretty well-performing models. To not take it for granted, _we now clearly state in the introduction_ that the relation between goodness and the uniformity of the Pareto front is an assumption that we hope to share with practitioners. Namely, we added the following text: "Throughout this work,
> we consider a good mapping one which uniformly maps our objective preferences (importance values going from zero to one) to the Pareto front, thus easing its navigation through our preferences."
>
> > The experiments [...] fail to provide independent evidence (e.g., downstream utility or expert agreement) that this uniformity actually translates to superior decision quality compared to baselines.
>
> In order to address the reviewer’s concerns and externally assess whether COPA better maps user preferences than existing methods, we plan to run an external user study. However, as this can be time consuming, we would appreciate it if the reviewer could share some feedback on what they would like to see in this study. We propose two alternatives, but are happy to hear other options:
>    1. We show users plots like those from Figure 1 without the method’s name, and directly ask which one reflects better their expectations.
>    2. We show a plot like those in Figure 1 with a colorless Pareto front, and ask the user to select one point for randomized preferences (values of alpha). Then, we compute the distance between those points and the ones selected by each method.
>
> > Additional discussion of when COPA is not appropriate (e.g., constraint-driven or regulatory settings) would help better position the scope of the method.
>
> We appreciate the reviewer’s feedback. Note that we already had a limitation section where we discussed settings for which COPA may not work (e.g. if two objectives are perfectly correlated). However, we have extended the limitation section with other cases and discussed palliative measures. We hope this helps better position our scope.

---

> > ### Comment · Reviewer_MHhY · 2026-04-07
> > **Remaining concerns about motivation and evaluation**
> >
> > Thank you for your detailed response and clarifications. I appreciate the effort to address my earlier concerns.
> >
> > That said, I still find the motivation, technical contribution, and practical usefulness of this work to be limited in its current form.
> >
> > In my view, the paper would need at least a few concrete examples, ideally realistic real-world cases, together with quantitative results, to more convincingly demonstrate both the practical motivation of the problem and the effectiveness of the proposed method. The readability of this paper should be improved.
> >
> > I would also encourage the use of more rigorous benchmarks and more standard evaluation metrics. The metric proposed in the paper can certainly be included as part of the analysis, but relying more heavily on widely accepted benchmarks and metrics would make the empirical conclusions easier for readers to interpret and trust. I still have questions about the experiments.

---

> > > ### Author Response · Authors · 2026-04-13
> > >
> > > Thank you for the follow up! We are happy to address any remaining concerns if they are actionable and clearly specified.
> > >
> > > As for what we can respond to, first please note that TMLR specifically aims to broaden the acceptance criteria beyond technical contributions and is therefore not a reason for rejection, see [here](https://jmlr.org/tmlr/acceptance-criteria.html). Regarding the experiments, all of them except the one used to ablate COPA are concrete and real examples taken from publicly-available leaderboards and publications. We acknowledge the reviewer’s desire to see quantitative results, but we are not aware of a rigorous benchmark nor standard metrics, and none have been suggested during this rebuttal. However, we offered in the rebuttal an option to provide quantitative results through a user study for which we are still waiting to know whether it would address the reviewer’s concerns and, if so, we are happy to carry out said user study.
> > >
> > > Please do not hesitate to share those remaining questions regarding our experiments (note that we submitted all the source code needed to reproduce them).

---

> > > > ### Comment · Reviewer_MHhY · 2026-04-13
> > > > **Concerns about the Experiments and Presentation**
> > > >
> > > > My concern is whether the proposed method provides benefits beyond the metric defined in this paper, and more importantly, whether it improves practical and widely accepted evaluation metrics, such as accuracy, recall, or other standard task-specific measures. If such results are already included in the paper, then these results should be clearly highlighted and discussed. If not, then I find it difficult to see the practical value of the proposed method, and the claimed usefulness of the approach is not convincing.
> > > >
> > > > In addition, my suggestion that the paper include more concrete examples was intended to improve its readability and to make its practical relevance clearer. At present, the paper is written in a highly theoretical manner, but it remains unclear what the method is actually useful for in practice. More realistic examples or application scenarios would help readers better understand why this method matters and in what setting it is expected to be useful.

---

### Review · Reviewer_oNmu · 2026-03-17

**Summary Of Contributions:**

This paper studies the problem of model selection under multiple, potentially incomparable objectives such as accuracy, fairness, and robustness. The authors propose COPA, a method that transforms heterogeneous objectives into comparable quantities using cumulative distribution functions derived from relative rankings. This allows aggregating multiple objectives while preserving their relative importance and enabling effective navigation of the Pareto front.

Strengths:
- Addresses an important and practical problem in ML model evaluation.
- The proposed method is simple, intuitive, and broadly applicable.
- Demonstrates utility across diverse domains including fair ML, AutoML, and foundation models.

Weaknesses:
- The methodological novelty is somewhat limited, as ranking-based normalization has prior precedents.
- Theoretical justification is relatively weak.
- Experimental comparisons could be strengthened with more rigorous baselines.

**Additional Comments:**

Overall, this is a well-written and practically motivated paper. While the core idea is relatively simple, it addresses an important gap in multi-objective model evaluation. Strengthening the theoretical grounding and empirical comparisons would significantly improve the impact of the work.

**Audience:**

Yes

**Audience Explanation:**

Yes. The problem of multi-objective model evaluation is highly relevant to the TMLR audience, especially given the increasing importance of fairness, robustness, and efficiency in modern ML systems. The proposed approach provides a practical tool that could be useful for both researchers and practitioners working on model selection and benchmarking.

The paper addresses a common and challenging problem in modern machine learning pipelines, where multiple objectives must be balanced. As ML systems become more complex and are deployed in real-world settings, practitioners increasingly need principled ways to compare models across heterogeneous metrics. Therefore, the proposed method is likely to attract interest from both academic researchers and industry practitioners.

**Broader Impact Concerns:**

The proposed method itself does not introduce significant ethical risks. However, since it can be used to balance fairness and performance metrics, it may influence decision-making in sensitive applications. It is important that practitioners carefully consider how preferences are specified and ensure that fairness-related objectives are not unintentionally deprioritized.

**Claims And Evidence:**

Yes

**Claims Explanation:**

The claims are generally supported by empirical results across several application domains, as described in the paper (see examples mentioned in the abstract on page 1 :contentReference[oaicite:0]{index=0}). The experiments demonstrate that COPA can effectively aggregate multiple objectives and assist in navigating Pareto fronts.

However, the evidence is primarily empirical and qualitative. The paper would benefit from more rigorous quantitative comparisons against strong multi-objective optimization baselines and clearer ablation studies to isolate the contribution of each component.

**Requested Changes:**

Critical:
- Provide stronger comparisons with established multi-objective optimization methods (e.g., scalarization, hypervolume-based methods, Pareto ranking).
- Include more quantitative evaluation to demonstrate consistent improvements over baselines.
- Clarify the theoretical properties of the proposed transformation (e.g., invariance, robustness, or consistency).

Non-critical:
- Improve clarity of the method description, especially the role of cumulative functions.
- Provide more intuition and visualizations to explain how COPA behaves under different distributions of objectives.
- Include computational complexity analysis.

---

> ### Author Response · Authors · 2026-03-31
>
> Thanks a lot for the review. We are happy to hear that our proposed methodology addresses an important practical problem while remaining simple and intuitive, and that our claims are empirically supported across several domains, demonstrating the utility of COPA.
> Let us next address the reviewer’s concerns:
>
> > Provide stronger comparisons with established multi-objective optimization methods (e.g., scalarization, hypervolume-based methods, Pareto ranking).
>
> Note that COPA is _not_ a MOO method, but a method for multi-objective evaluation. As such, methods like hypervolume-based methods address a different problem, namely, finding the Pareto front, whereas we assume a given Pareto front (first line of section 2). Also note that we compare and discuss similar methods (SAW, AHP, MEW, or FUCA) in detail in appendix A.
>
> > The paper would benefit from [...] clearer ablation studies to isolate the contribution of each component.
>
> We would like to point out the ablation studies that one can find in the appendix. Specifically, we explicitly check the robustness of the estimator to sample size in App. B.1.1, and we show the difference between the proposed norm and a usual weighted $p$-norm in App. B.2.1. We are happy to run more ablations the reviewer could consider necessary, as we did for reviewer PZuw (see Figure B.2)
>
> > Include more quantitative evaluation to demonstrate consistent improvements over baselines.
>
> We want to stress that quantifying the degree to which we map user preferences to the Pareto front is challenging. Instead, we provide an extensive set of experiments that qualitatively demonstrate our claims in a wide range of use cases. In addition, we plan to run a user study to externally verify our results. Could the reviewer share whether this would help easing their concerns and, if so, _what questions would they like users to answer_? Please check the response to reviewer MHhY for further details.
>
> > Clarify the theoretical properties of the proposed transformation (e.g., invariance, robustness, or consistency).
>
> We are happy to clarify specific properties that the reviewer considers require clarification. Please note that we devote section 3.2 to discuss the properties of our estimator: we characterize its mean and variance in Prop. 3.1, and state its consistency and uniform convergence right after. Besides, since it is order-preserving (D4), it is invariant to bijective transformations.
>
> > Improve clarity of the method description, especially the role of cumulative functions.
>
> In order to clarify the method description, it would help us a lot to understand what exactly is not clear at the moment. Regarding the role of cumulative functions, we clearly state (section 3.1, Eq. 8) that CDFs are our normalization functions, which we approximate using samples (section 3.2, Eq. 9).
>
> > Include computational complexity analysis.
>
> We appreciate the feedback! In short, COPA needs to: 1) sort the samples of each task (Eq. 9), and 2) compute a weighted norm (Eq. 11). As a result, COPA’s complexity is that of sorting K vectors of size N, $\mathcal{O}( K N \log N)$ where $K$ is the number of objectives and $N$ the number of samples. We have clarified this in the updated version (page 7, before the related work).
>
> > However, since it can be used to balance fairness and performance metrics, it may influence decision-making in sensitive applications.
>
> We fully agree with the reviewer. However, we would like to also bring attention to the following: Since incomparability is not being currently addressed, _existing decision making is already affected by it_ which, as we show in our fairness use-case (Figure B.4.a) it can bias decisions towards a single model.

---

### Decision · Action_Editor_fPAo · 2026-04-17

**Recommendation:** Reject

**Additional Comments:**

I have decided to reject this submission, primarily aligning with the concerns raised by reviewer MHhY. While the problem of multi-objective model evaluation is certainly of interest to the TMLR audience, the claims made in the submission regarding the method's practical effectiveness are not adequately supported by clear and convincing evidence. The current empirical evaluation is insufficient. I suggest that the authors include more rigorous benchmarks and widely accepted evaluation metrics, such as accuracy and recall, to demonstrate the method's practical value and effectiveness beyond their own defined metrics. The authors are encouraged to incorporate this feedback and consider submitting the revised work to a future venue.

**Audience:**

Yes

**Audience Explanation:**

The paper addresses heterogeneous multi-objective evaluation, which is a practical and widely encountered problem in ML. The demonstration of how normalization and aggregation choices can substantially influence rankings and interpretations in model evaluation and benchmarking is highly relevant to the TMLR audience.

**Claims And Evidence:**

No

**Claims Explanation:**

While the paper provides empirical demonstrations, the evaluation does not include standard and widely recognized metrics, such as accuracy, recall, or other task-specific measures, for a fair and objective assessment. The experiments evaluate the method primarily using the authors' own proposed metric, making it difficult to assess its practical effectiveness or compare it meaningfully with existing approaches.

**Resubmission Of Major Revision:**

The authors may consider submitting a major revision at a later time.